# A microengineered vascularized bleeding model that integrates the principal components of hemostasis

Yumiko Sakurai[1,2], Elaissa T. Hardy[1,2], Byungwook Ahn[1,2], Reginald Tran[1,2], Meredith E. Fay[1,2], Jordan C. Ciciliano [3], Robert G. Mannino[1,2], David R. Myers[1,2], Yongzhi Qiu[1,2], Marcus A. Carden[2], W. Hunter Baldwin[2], Shannon L. Meeks[2], Gary E. Gilbert[4], Shawn M. Jobe[5] & Wilbur A. Lam[1,2]

Hemostasis encompasses an ensemble of interactions among platelets, coagulation factors, blood cells, endothelium, and hemodynamic forces, but current assays assess only isolated aspects of this complex process. Accordingly, here we develop a comprehensive in vitro mechanical injury bleeding model comprising an "endothelialized" microfluidic system coupled with a microengineered pneumatic valve that induces a vascular "injury". With perfusion of whole blood, hemostatic plug formation is visualized and "in vitro bleeding time" is measured. We investigate the interaction of different components of hemostasis, gaining insight into several unresolved hematologic issues. Specifically, we visualize and quantitatively demonstrate: the effect of anti-platelet agent on clot contraction and hemostatic plug formation, that von Willebrand factor is essential for hemostasis at high shear, that hemophilia A blood confers unstable hemostatic plug formation and altered fibrin architecture, and the importance of endothelial phosphatidylserine in hemostasis. These results establish the versatility and clinical utility of our microfluidic bleeding model.

[1] Wallace H. Coulter Department of Biomedical Engineering, Georgia Institute of Technology and Emory University, 345 Ferst Drive, Atlanta, GA 30332, USA. [2] Department of Pediatrics, Divisoin of Pediatric Hematology/Oncology, Aflac Cancer Center and Blood Disorders Center of Children's Healthcare of Atlanta, Emory University School of Medicine, 2015 Uppergate Drive, Atlanta, GA 30322, USA. [3] George W. Woodruff School of Mechanical Engineering, Georgia Institute of Technology, 801 Ferst Drive, Atlanta, GA 30332, USA. [4] Medicine Depts of VA Boston Healthcare System and Harvard Medical School, 150 S Huntington Ave, Boston, Massachusetts 02130, USA. [5] Blood Center of Wisconsin, 8733W Watertown Plank Road, Wauwatosa, WI 53226, USA. Yumiko Sakurai, Elaissa T. Hardy and Byungwook Ahn contributed equally to this work. Correspondence and requests for materials should be addressed to S.M.J. (email: Shawn.Jobe@BCW.edu) or to W.A.L. (email: wilbur.lam@emory.edu)

Following vascular injury, the hemostatic response is activated and a complex, yet carefully balanced, ensemble of biological, biochemical, and biophysical interactions is initiated[1, 2]. Under the hemodynamic shear conditions of the circulation, platelets initially adhere to the vascular wound site via von Willebrand factor (vWF) and collagen binding. Adherent platelets release biochemical agonists that induce platelet aggregation and result in the formation of a hemostatic plug, thereby triggering the coagulation cascade to initiate fibrin polymerization and establish a more stable clot[3]. While in vitro assays have enabled significant advances in our understanding of this complicated process, the currently available hemostasis assays only assess isolated aspects of clot formation, which has stymied the fields of clinical and experimental hematology given the interdependence of the various components of hemostasis.

Current bleeding tests are restricted to isolated analysis of components of coagulation (e.g., prothrombin time, activated partial thromboplastin time), vWF, or platelet function (e.g., platelet function analyzer and aggregometry)[4–6]. Even more "global" hemostasis assays, such as thromboelastography and thrombin generation-based tests, fail to take into account the role of either the endothelium or shear stress. While in vivo animal injury models have enabled groundbreaking research in hemostasis, data obtained from these non-primate models may not directly translate to human physiology and disease[7–9].

Recent advances in microfabrication technologies have provided useful, inexpensive, and easily reproducible microfluidic platforms for conducting clinically relevant, microscale biological and biochemical experiments. Accordingly, numerous research groups, including our own, have recently applied microfluidic devices to study hemostasis and thrombosis[10–15]. However, these microdevices assay clot formation via perfusion of blood over surfaces patterned with clot-activating substances—collagen, kaolin, or tissue factor (TF)—leading to accumulation of platelet aggregates and fibrin[16,] and therefore function more as models of thrombosis rather than hemostasis, where hemostasis is specifically defined as cessation of bleeding after vascular injury. While Schoeman et al. recently presented an elegant microfluidic system that probes bleeding time using a collagen/TF-coated microchannel[17], it incorporates neither the limiting anticoagulant effect of intact endothelium nor the potential procoagulant activity of injured endothelial cells[18]. Therefore, a clear need exists for an in vitro model of the hemostatic response that integrates all of the major biological, biochemical, and biophysical components of hemostasis in the context of vascular injury.

To that end, we have developed an in vitro vascularized microfluidic mechanical injury bleeding model by integrating the fabrication of a pneumatic microvalve and "endothelialization" of microfluidics[12, 13, 19–21]. Our microfluidic uniquely comprises a completely "endothelialized" microchannel to serve as a vasculature model and a pneumatic microvalve functions as a "trap door" to enable positive pressure to mechanically disrupt, and therefore injure, the vascularized microchannel resulting in "bleeding" into a separate microchannel. The anti-thrombotic properties of the live in vitro endothelium enables the use of human whole blood minimally anticoagulated with corn trypsin inhibitor (CTI) to inhibit the contact pathway of clotting as blood initially traverses the syringe/tubing before entering the microfluidic system. Here we demonstrate that this in vitro microsystem enables direct, real-time visualization of the entire hemostatic process upon mechanical vascular injury with single-cell resolution, while allowing for tight control and modulation of the major cellular (e.g., endothelial cell type, inclusion/exclusion of blood cell subpopulations); biomolecular (e.g., pharmacological agents, inhibitory antibodies); and biophysical (e.g., shear stress) components of hemostasis. In addition, cessation of "bleeding" in

our system can be directly measured, resulting in an in vitro analog of the bleeding time. To demonstrate the versatility of this model, we investigate key questions within the field of hemostasis/thrombosis that have previously been technologically infeasible to address directly. Specifically, we show that: (a) under physiologic flow conditions, the anti-platelet agent eptifibatide predominantly affects hemostatic plug formation via attenuation of platelet aggregate density and clot contraction ; (b) inhibition of vWF prolongs the in vitro bleeding time in a shear-dependent manner; (c) phosphatidylserine (PSer) is expressed primarily on the surface of endothelial cells at the site of vascular injury; and (d) blood from hemophilia A patients features abnormal fibrin architecture and significantly increased bleeding times. Overall, these data establish the research-enabling and hypothesis-generating versatility of our vascularized microfluidic mechanical injury bleeding model for studying human blood samples and its promise as a drug discovery platform for hemostatic and thrombotic disorders.

## Results

**An in vitro mechanical injury bleeding and hemostasis model**. The polydimethylsiloxane (PDMS)-based microfluidic bleeding time device (Fig. 1a) is comprised of three layers: a top "vascular" layer that features an endothelialized microchannel (vascular channel) that recapitulates the microvasculature, a middle valve layer that comprises a PDMS membrane the position of which can be pressure controlled, and a bottom valve actuator layer that allows pressure to be introduced to the system to control valve position (Fig. 1b). To create the endothelialized vascular channel, the microsystem is precoated with collagen type 1 and fibronectin, seeded with human umbilical cord vein or aortic endothelial cells, and then cultured under physiologic flow conditions until the cells reach confluence—the valve is closed throughout this process (Fig. 1c, Valve closed). To assess the hemostatic response, vascular "injury" is introduced by providing positive fluidic pressure through and into the vascular channel and layer, and simultaneous negative pressure in the "valve actuator" layer to effectively pull down the valve a distance of 6 μm. This disrupts the endothelium, creating a "bleed" with a newly formed wound channel now continuous with the vascular channel (Fig. 1c, Valve open, Wound created). The valve is then maintained in the open position, with the outlet channel at ambient pressure. Immediately after injury, re-calcified CTI-treated whole blood is perfused through the device at a controlled flow rate (Fig. 1c, Valve open, Bleeding). Real-time monitoring of hemostasis can then be performed via microscopy. Wound width, bleeding time (defined as the time at which red blood cells stop transiting into the wound area), and fluorescence intensity of relevant biological species are then monitored. Importantly, in the absence of endothelial injury, no platelet adherence, activation, or clot formation is detected upon perfusion of the vascular channel with whole blood.

**Characterization of the microfluidic bleeding device**. To investigate how biophysical and biochemical variables interact to mediate the hemostatic response, we first characterized various aspects of bleeding and hemostasis in our "control" conditions: confluent human umbilical vein endothelial cells (HUVECs) and re-calcified blood treated with CTI, anti-CD41 to fluorescently label platelets, and fluorescently labeled fibrinogen. The width of the mechanically induced wound varies between experiments as it is a result of the valve size and the user applied pressure. Across 47 experiments, we found the mean width to be $132.49 \pm 40.19$ μm (range of minimum 73.22 to maximum 231.86 μm). To more fully describe the fluidic microenvironment, computational fluid dynamics simulations were performed, which showed that, for the

average wound size, the initial shear stress at the middle of the wound site is approximately 3 dyne cm$^{-2}$ (range: 2–5 dyne cm$^{-2}$) and at the edge of wound approximately 7 dyne cm$^{-2}$ (range: 4–8 dyne cm$^{-2}$) when whole blood was perfused at 500 s$^{-1}$ (Fig. 1d). Analysis of the shear gradient demonstrates that the highest shear

occurs at the injury site (Supplementary Fig. 1). The pressure drop along the straight section of the vascular channel was also simulated with and without injury, yielding pressure drops on the order of 10$^2$ Pa. We determined that the addition of the injury negligibly reduced the pressure drop by 2.7%, which is consistent

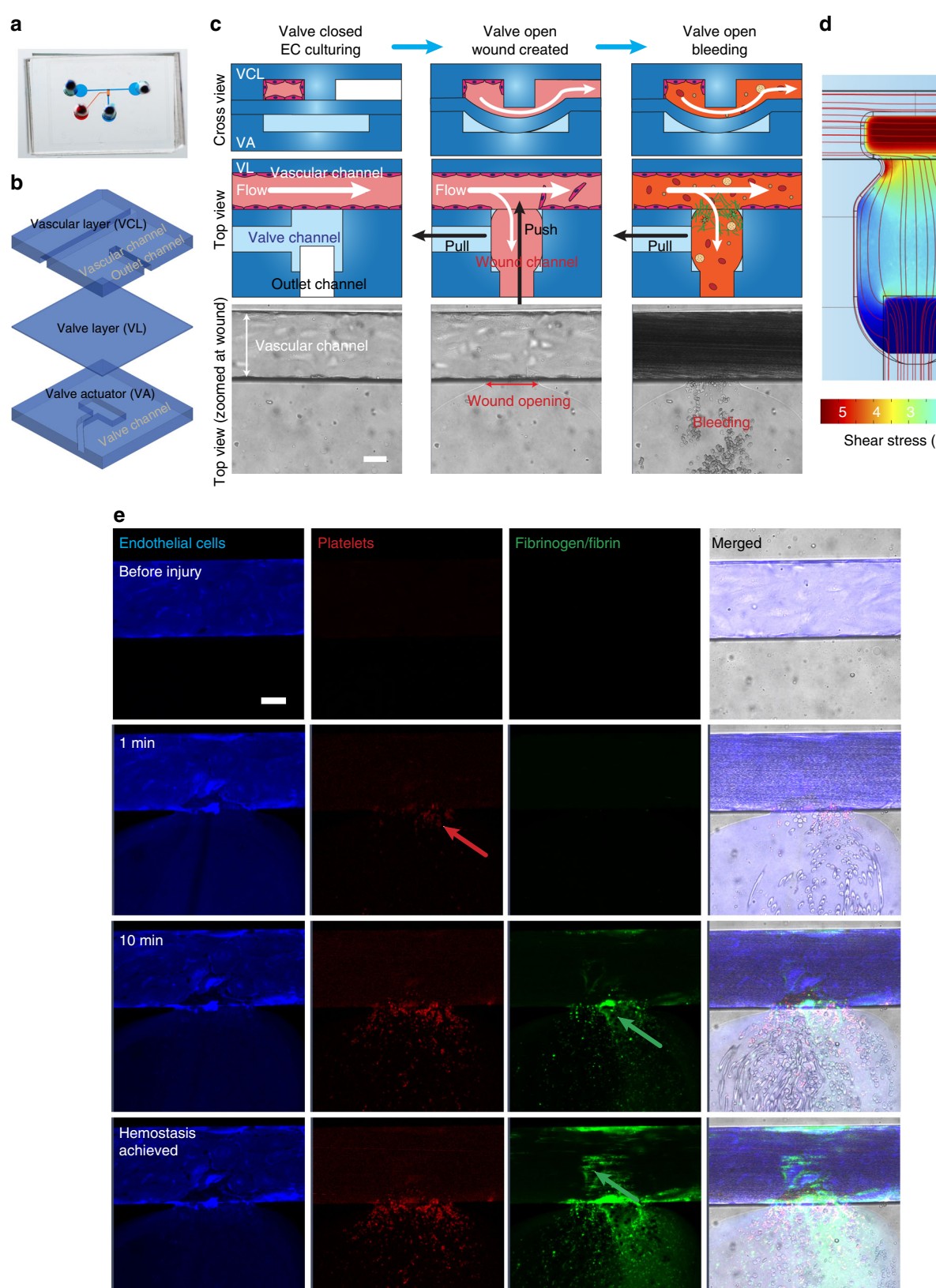

with hydraulic circuit analysis calculations[17]. As expected, upon mechanical introduction of the wound, blood not only continues to perfuse the vascular channel (Fig. 1c, horizontal flow) but also flows into the newly formed wound channel, analogous to "bleeding" (Fig. 1c, vertical flow). As blood continues to flow through the vascular and wound channels, platelets adhere throughout the injury site, platelet aggregation leads to the formation of a hemostatic plug at the wound site, fibrin polymerization occurs, and finally, cessation of blood flow is achieved (Fig. 1e, Supplementary Movie 1). Experimentally, median time to cessation of blood flow into the wound channel (i.e. the "bleeding time") for healthy donors was 503 s. We observed that the initial wound width and bleeding time did not correlate in our studies (Supplementary Fig. 2a, $R^2 = 0.088$), which is likely due to the fact that the shear stress profiles are all within the same order of magnitude over the operable range of wound widths (Supplementary Fig. 2b). Therefore, in our system, the time course of hemostasis is not dependent on initial wound width or shear stress. We posit that this is in part due to platelets immediately adhering at the wound site and reducing the wound opening. Interestingly, compared to venous conditions, platelet adhesion, fibrin accumulation, and bleeding time in our system were similar under arterial cellular and shear conditions, when human aortic endothelial cells (HAECs) were seeded in the vascular microchannels and blood was perfused at the average aortic shear rate of 2500 s$^{-1}$ (five times higher than that utilized in our studies using HUVECs) (Supplementary Fig. 3). Conversely, in the absence of endothelial cells, platelets adhered to the collagen-coated microchannels, but hemostasis was not achieved within the maximum recording time (1200 s, $n = 3$ experiments). This suggests that the injured endothelial cells, in and of themselves, serve a procoagulant function that is required for hemostasis when there is no exogenous source of TF. We also visually confirmed the expression of P-selectin, a marker of inflammation, via antibody binding, on the surfaces of platelets over the time course of hemostasis (Supplementary Fig. 4).

**Visualizing hemostasis under altered microenvironments**. Our microfluidic was then examined in the context of hematologically altered conditions. A potential side effect of integrin $\alpha_{IIb}\beta_3$ antagonism in the prevention of arterial thrombosis is bleeding. To investigate how this affects hemostatic plug formation, healthy human blood was treated with 10 μg ml$^{-1}$ eptifibatide. There was no significant difference in bleeding time between eptifibatide-treated and vehicle control blood samples (Fig. 2a). Interestingly, while there was no difference in the total area of adherent platelets between the two conditions (Fig. 2b), platelet aggregation and contraction of those aggregates, as shown via saturated platelet fluorescence, was more pronounced in control blood than in eptifibatide-treated blood (Fig. 2c, d and Supplementary Movie 2). Also, by visualizing subpopulations of platelets fluorescently labeled with a different color (calcein, in green) we

confirmed clot contraction (i.e., platelets in the clots are visibly displaced proximally and against the direction of flow) in healthy control blood, but this phenomenon was attenuated in eptifibatide-treated blood (Supplementary Movies 3 and 4). Higher magnification visualization of the bleeding site revealed clear co-localization of fibrin(ogen) with platelets in control conditions but not in the eptifibatide-treated condition (Fig. 2e), and this was demonstrated with Pearson correlation coefficient analysis (Fig. 2f–h). Overall, these results suggest that the integrin $\alpha_{IIb}\beta_3$–fibrin interaction "glues" platelets together at the site of vascular injury, but that this interaction alone is not necessary for hemostasis in the microvasculature.

vWF is a shear-responsive protein essential for platelet adhesion in conditions of arterial flow[22]. Moreover, the interaction of factor VIII with vWF allows the persistence of circulating factor VIII within the vasculature. vWF co-localized wth endothelial cells at the wound site and in platelets in the hemostatic plug (Fig. 3a). The AVW3 antibody blocks the interaction of vWF with its cognate platelet receptor GPIb-IX without affecting the interaction of vWF with factor VIII. Using AVW3, we quantitatively assessed how inhibition of the GPIb–vWF interaction specifically affects hemostatic plug formation. The hemostatic response using AVW3-treated whole blood was assessed at venous (500 s$^{-1}$) and arteriolar (2500 s$^{-1}$) shear rates. Strikingly, yet consistent with the known shear-responsive nature of vWF, vWF inhibition almost exclusively prevented hemostasis in conditions of arteriolar flow. At a shear rate of 500 s$^{-1}$, the median bleeding time in the presence of AVW3 was 559.5 s, which was not statistically different compared to that of control conditions, while at 2500 s$^{-1}$, hemostasis was not achieved over the experimental time scales in the presence of AVW3 (Fig. 3b), indicating that a specific deficiency of vWF activity primarily impacts hemostasis at high shear.

PSer is an integral component of the nascent hemostatic plug, as its exposure on cell surfaces provides an essential surface for the assembly of coagulation enzyme complexes and the amplification of the coagulation cascade[23]. The timing and localization of PSer exposure was examined using fluorescently labeled annexin V (Fig. 4). Importantly, confluent endothelial cells did not display PSer prior to injury (Fig. 4, left top). Blood was perfused at 500 s$^{-1}$ (Supplementary Movies 5 and 6), and immediately upon wound initiation, PSer was detected on endothelial cells, but only in the vicinity of the wound area. The amount of PSer exposure on the injured endothelial cells increased over time, while the plasma membrane stain decreased; this trend continued locally even after hemostasis was achieved. Injured cells were readily discerned even at the initial time of wounding by the disruption of previously confluent cell-cell contacts. PSer-positive (i.e., PSer is exposed on the surface of platelet) events could be observed both at the sites of denuded endothelium adjacent to the intact endothelium and within the hemostatic plug. However, the majority of these platelets were

**Fig. 1** An "endothelialized" microfluidic system coupled with a microengineered pneumatic valve that induces a vascular "injury" functions as a comprehensive in vitro mechanical injury bleeding model. **a** Our microfluidic-based bleeding device is constructed from **b** three polydimethylsiloxane layers: a vascular layer comprised of a vascular channel and outlet channel, a valve layer, and a valve actuator layer. **c** First, endothelial cells are cultured to confluence in the vascular channel while the pneumatic valve is in the closed position (left). When the valve layer is pulled down by negative pressure in valve actuator channel (pull) and mechanical fluidic pressure (push) is applied through outlet channel, the vascular channel is disrupted and endothelial injury is mechanically induced (middle). Blood is perfused while the valve is maintained in the open position and flows through the vascular channel (horizontal) as well as the newly created "bleeding" wound channel, which leads to the outlet channel (right). **d** A computational fluid dynamic COMSOL simulation shows initial wall shear stresses when the mechanical injury is introduced. **e** Our in vitro microcvasculature mechanical injury model enables real-time visualization of the entire hemostatic process. Confluent endothelial cells (stained via a plasma membrane stain—blue) are disrupted when the valve opens and "bleeding" occurs through the injury site. Immediately, platelets (stained via CD41—red) begin to adhere at the injury site (red arrow) followed by fibrinogen/fibrin accumulation (fluorescent tagged fibrinogen, green) until finally hemostasis is achieved. All scale bars = 50 μm

PSer negative (i.e., PSer was bound to the inner leaflet of the platelet membrane and not exposed to the surface, and therefore not detected with the annexin V stain). Also when we used fibrin-specific antibody 59D8[24], we observed significant amounts of fibrin accumulation at the damaged edges and detached portions of the endothelial cell membranes (Supplementary Fig. 5). This suggests that PSer expressed on injured endothelial cells serves as the principal prothrombotic surface for initial hemostasis, while platelet PSer does not appear to play a significant role initially. These studies highlight our system's capabilities for controlled

visual analysis of the many interconnected facets of hemostasis, and provides spatiotemporal insight into these phenomena.

TF, a ligand for factor VIIa that leads to thrombin production and clot formation, is an integral component in the initiation of the coagulation cascade both in vivo and in vitro[17]. In vivo, TF is contained within the subendothelium and smooth muscle cells, with another potential circulating source in white blood cells, which can be observed within the hemostatic plug (Supplementary Fig. 6a and Supplementary Movie 7). To investigate whether TF stemming from leukocytes or other cell sources was required for hemostasis in our system, we used whole blood with an anti-TF

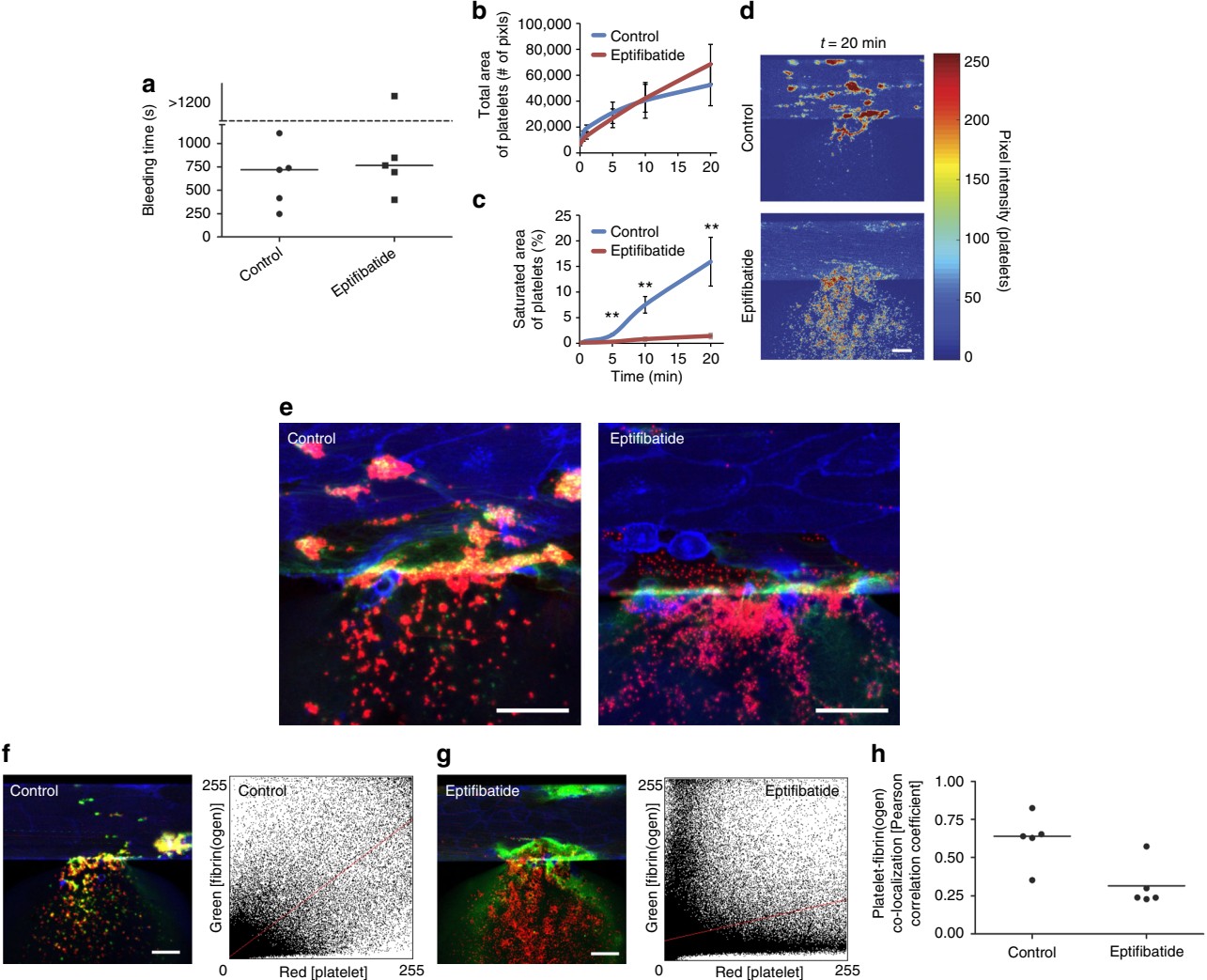

**Fig. 2** Integrin $\alpha_{IIb}\beta_3$ antagonism alters hemostatic plug architecture by inhibiting platelet–fibrin binding and clot contraction. **a** Healthy blood treated with 10 µg ml$^{-1}$ eptifibatide, an anti-platelet drug commonly used in the clinical setting, and healthy blood with vehicle control exhibit similar in vitro bleeding times. **b** Moreover, the total areas of adhered platelets are also similar between the two conditions. **c** However, when comparing platelet aggregation followed by clot contraction, which is tracked by measuring the areas of saturated intensity of fluorescently labeled platelets over time, eptifibatide-treated blood exhibited significantly less platelet aggregation and clot contraction over the time course of hemostasis as compared to vehicle control conditions. The error bars in the graphs show standard errors (control $n = 3$, eptifibatide $n = 4$). Double asterisk (**) shows P-value < 0.05 by Student's t-test. **d** Heat map of adhered platelets shows dense platelet aggregates with saturated fluorescence intensity formed in vehicle control conditions at 20 min after bleeding (top) while no aggregation or clot contraction observed in eptifibatide-treated blood (bottom). **e** Under the eptifibatide-treated conditions, platelet–fibrin binding (as defined by co-localization of platelets—red—and fibrinogen/fibrin—green) was markedly decreased compared to that of the control conditions. **f**, **g** The analyzed images of a representative experiment with healthy control and eptifibatide-treated blood samples (left) and the corresponding platelet-fibrin(ogen) co-localization plot (right). Each co-localization plot displays the pixel intensity of the red channel (x axis), corresponding to platelets, plotted against the pixel intensity of the green channel (y axis), corresponding to fibrin(ogen), of each pixel in the respective image. The red lines show the Pearson correlation coefficient lines. **h** Comparing Pearson correlation coefficients between healthy control and eptifibatide-treated samples ($n = 5$ for each condition) reveals that co-localization is more pronounced in the control condition (Student's t-test, $P = 0.008$). All scale bars = 50 µm. Bars in graph represent median values

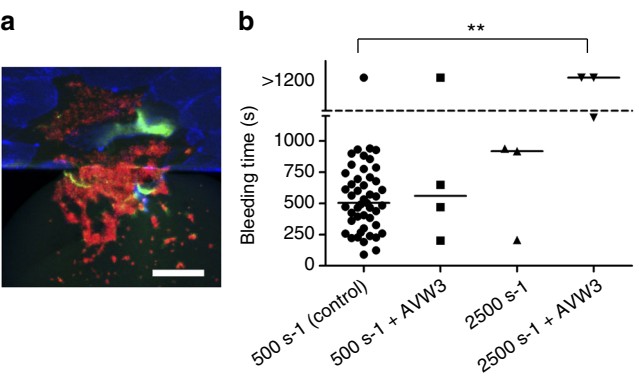

**Fig. 3** Functional inhibition of vWF causes prolonged bleeding in high shear conditions. **a** vWF (green) co-localizes with injured endothelial cells as well as with platelets. Endothelial cells are stained with plasma membrane stain (blue) while platelets are stained with anti-CD41 (red). **b** When healthy blood was treated with the AVW3 antibody, which binds and inhibits the A1 domain of vWF, we observed prolonged bleeding time only at arteriolar ($2500 \text{ s}^{-1}$) shear rates comparted to venular shear rates ($500 \text{ s}^{-1}$). One-way ANOVA indicate statistically significant difference among these four conditions, and Dunn's multiple comparison test further show that bleeding times are significantly different between the $500 \text{ s}^{-1}$ control and $2500 \text{ s}^{-1}$ with AVW3 conditions (**). Scale bar = 50 μm. Bars in graph represent median values

antibody (TF9-10H10). There was no difference in bleeding time in TF-inhibited samples as compared to controls ($n = 2$). This suggests that injured endothelial cells and exposed collagen are sufficient to elicit a clotting response in the context of whole blood and physiologic shear stress. To assess the effect of exogenous TF in our system, vascular microchannels were precoated with 1 nM of TF and collagen prior to the seeding endothelial cells. We observed enhanced fibrin(ogen) accumulation at early time points (Supplementary Fig. 6b and c). These preliminary results highlight the utility of our system in assessing the synergistic relationship between coagulation factors in the hemostatic process.

To demonstrate the clinical relevance of the in vitro bleeding time in our microfluidic in the context of coagulation disorders, healthy blood was treated with an antibody against the A2 domain of Factor VIII (anti-A2 mAb 2–76) to induce a severe hemophilia A phenotype. In the absence of functional factor VIII, fibrin accumulation was minimal over the experimental time course. While platelet accumulation was unaffected (Fig. 5a), significant reduction of fibrinogen/fibrin was observed at later time points (Fig. 5b, at 10 and 15 min), and this absence of coagulation resulted in an unstable clot with repeated re-bleeding (Fig. 5c and Supplementary Movie 8). In fact, hemostasis was not achieved in any of the antibody-induced hemophilia samples (Fig. 5d, $n = 8$). Furthermore, we obtained clinical blood samples from severe Hemophilia A patients and found that hemostasis was not achieved in our microfluidic bleeding model for any of the patient hemophilia samples (Fig. 5e, $n = 4$, Supplementary Movie 9). As expected, we again saw a clear reduction in fibrinogen/fibrin accumulation (Fig. 5f).

Taken together, these studies highlight how our microsystem provides spatiotemporal insight into the cellular and molecular interactions of hemostasis. Furthermore, by introducing clinical patient samples, our bleeding device allows us to study hemostasis in different disease states.

## Discussion

Here we describe the development and characterization of what, to our knowledge, is the first reported microfluidic-based in vitro

bleeding model that recapitulates key aspects of in vivo mechanical injury of the microvasculature. The microsystem uniquely incorporates the major aspects of hemostasis, including platelets (and all other blood cell types and subpopulations), coagulation factors, an intact and functional endothelium, and shear stress, all of which can be tightly controlled, included/ excluded, and varied. When used in conjunction with fluorescently labeled antibodies or cellular dyes, the entire hemostatic process can be measured and quantitatively evaluated in real time with single-cell resolution via microscopy.

The use of microfluidic platforms to study clot formation under physiologic flow conditions is well established, and their utility in differentiating limited aspects of the hemostatic process in both normal and pathophysiologic states has been elegantly demonstrated[17]. However, the presented system uniquely recapitulates important in vivo aspects of hemostasis after vascular injury not found in other microfluidic systems. First, blood vessel injury with blood loss due to mechanical trauma is accurately modeled by the endothelialized microchannel coupled with a microengineered pneumatic valve. This enables the visualization and quantitative assessment of hemostatic plug formation and time to cessation of "bleeding". In addition, the inclusion of the endothelium, with its antithrombotic properties, enables the use of only minimal anticoagulation with CTI, and therefore the inclusion of whole blood. This, in turn, uniquely allows for assessment of endothelial interactions with all blood cell sub-populations at the site of vascular injury under physiologic, or near physiologic, conditions. Moreover, we find that the injured endothelium, in concert with exposed collagen, is sufficient to cause hemostasis in the absence of TF.

Our microsystem allows us to make unique quantitative spatiotemporal insights into how, under conditions of flow and in the context of an intact and functional endothelium, components of clot formation function both physiologically and in pathologic states to initially re-establish vascular integrity. Eptifibatide, despite its clinically problematic effects on hemostasis, had no effect on in vivo bleeding time in both preclinical animal and clinical human trials[25, 26]. Consistent with these studies, we demonstrate that eptifibatide does not effect in vitro bleeding time. But here, through direct visualization of the hemostatic process, we identify that inhibition of the fibrin(ogen)–$\alpha_{IIb}\beta_3$ interaction dramatically alters the cellular architecture of human hemostatic plug formation. Specifically, in our system, while eptifibatide does not affect the number of platelets adhered to the collagen matrix, it leads to decreased clot contraction and therefore a lower density of platelets within the hemostatic plug (Fig. 2 and Supplementary Movies 2–4). As clot contraction is dependent on flow conditions[27], these data showcase the experimental versatility of our system, in this case in providing new physiologic insight into the effects of a well-established pharmacologic agent.

While vWF is known to be a shear-responsive protein[28], this effect has not been apparent in murine models of venous and arterial thrombosis. Here the capability to tightly control the hemodynamic environment in the context of vascular injury demonstrates that bleeding time is vWF dependent at high but not low shear conditions (Fig. 3) through the limitation of platelet accumulation, and that this effect can be separated from vWF's role as a carrier of factor VIII. On the other hand, severe deficiency of factor VIII activity severely limited hemostasis by preventing fibrin formation and the formation of a stable hemostatic plug. Together, these results demonstrate the ability of our microfluidic to differentiate the precise physiologic mechanism by which a hemorrhagic diathesis might occur.

Finally, this microsystem affords a reductionist approach that allows for the resolution of longstanding questions that have been

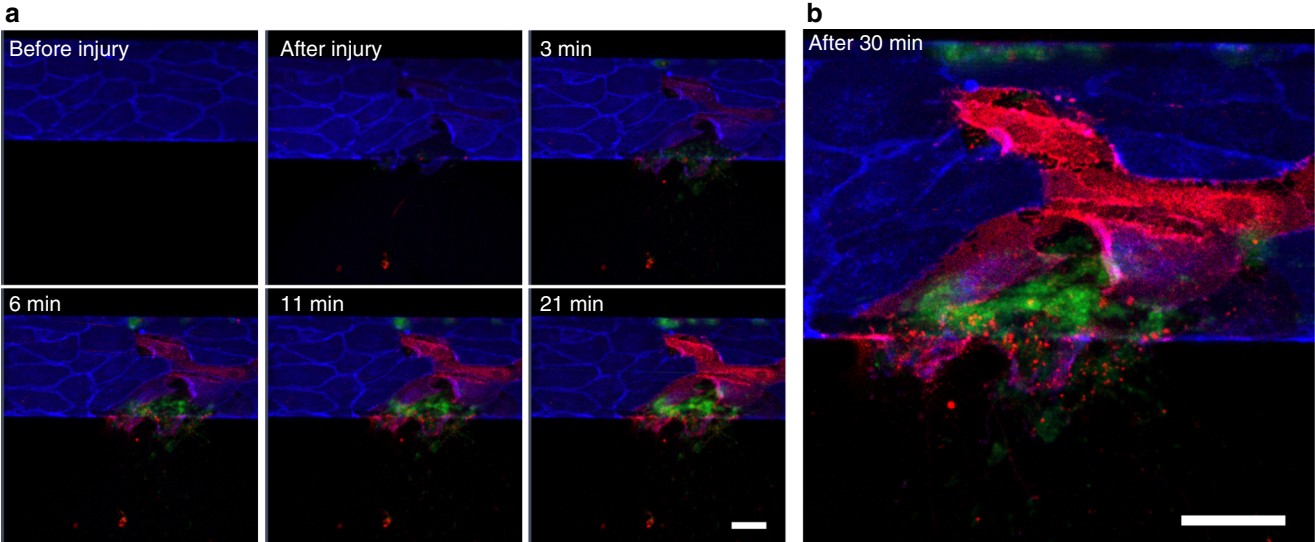

**Fig. 4** Annexin V binding reveals rapid exposure of phosphatidylserine (PSer) on endothelial cell surfaces but not on platelets. **a** Confluent endothelial cells before injury show no PSer exposure on their surfaces (top left). Immediately after mechanical injury is induced, the endothelial cells in the strict vicinity of the wound exhibit PSer exposure, as measured with annexin V binding (red) in a time-dependent manner while losing their plasma membrane staining over the time course (blue). In contrast, platelets (green) adhered at the injury site did not exhibit PSer exposure. **b** Thirty minutes after the injury was induced and hemostasis was achieved, the majority of the platelet aggregates did not exhibit significant amounts of PSer exposure, in contrast with endothelial cells at the wound site, which exhibit significant PSer exposure. All scale bars = 50 μm

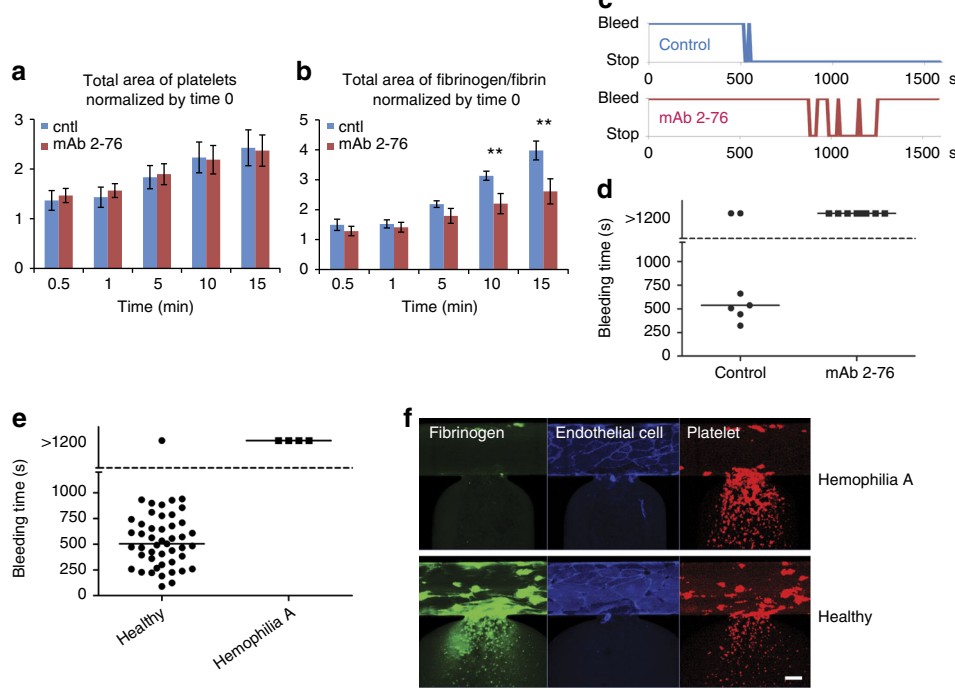

**Fig. 5** Inhibition of factor VIII and blood from hemophilia A patients reduce fibrinogen/fibrin accumulation at the injury site and impair hemostasis. **a** When healthy blood was treated with an antibody against A2 domain of Factor VIII (mAb 2–76), there was no difference in the total surface area of adhered platelets at the injury site compared to that of vehicle control conditions. **b** However, inhibition of factor VIII resulted in a significant reduction of fibrinogen/fibrin accumulation at later time points (10 and 15 min after the mechanical injury). The error bars in the graphs show standard errors (control $n$ = 4, mAb 2–76 $n$ = 4). Double asterisk (**) shows $P$-value < 0.05 by Student's $t$-test. **c** Antibody-treated blood sample exhibited "re-bleeding" and failed to achieve hemostasis during the entire experimental time frame (**d**). **e** Severe hemophilia A patient blood samples also failed to achieve hemostasis and continued bleeding over the entire experimental time course. **f** Fibrinogen/fibrin accumulation (green) was also markedly decreased in severe hemophilia A patient blood samples at the wound site as compared to that of healthy control blood, although platelet (red) adhesion and accumulation were similar between the two conditions. Scale bar = 50 μm. Bars in graphs represent median values

difficult to assess in vivo. For example, it has been debated whether the PSer exposure that contributes to fibrin formation at the site of vascular injury occurs first on the endothelial surface or the platelet surface. In our system, we clearly demonstrate that PSer is exposed on injured endothelial cells within seconds of a mechanical vascular injury and expression continues to increase in the direct vicinity of the injury over the time course of hemostasis, while platelet PSer expression at the wound site occurs later. While PSer exposure on endothelial cells has been reported as being more significant than platelet PSer, the timing of these different events has been unresolved and debated[18, 29–34]. This highlights the research-enabling aspect of our technology to generate new hypotheses.

No in vitro device can fully recapitulate all in vivo conditions, and our system is not without limitations. Owing to its small scale, our microsystem physically models the microvasculature rather than larger vessels that are dictated by different fluid and mass transport phenomena due to the larger characteristic lengths and flow rates observed in larger vessels. The microscale dimensions of our system result in steady,laminar flow, which enables shear stress to be tightly controlled and modulated to mimic venular/small venous to arteriolar/small arterial conditions, thereby enabling the dynamic physiologic range of the microcirculation[7]. We observe that the bleeding time remains unchanged under arterial and venous shear conditions with HAECs and HUVECs. We conjecture that under these different conditions, our system, in which there is a mechanical injury of relatively constant size that is induced at a specific site of an endothelialized channel, two opposing phenomenon are occurring. On the one hand, the increased pressure and velocity under arterial conditions should lead to more "bleeding" in our system. On the other hand, the increased flux of hemostatic components including platelets and coagulation factors should lead to a more rapid formation of a hemostatic plug, especially given the fact that the injury size is unchanged. This then diverts blood flow back to the vascular channel and additional platelet/fibrin accumulation ceases, which is what occurs physiologically. In addition, the endothelial cells themselves, which are from venous and arterial vascular beds, may also be biologically more "tuned" to their corresponding shear conditions. These phenomenon therefore likely collectively cancel each other out leading to similar hemostatic plug formation and bleeding times under arterial and venous conditions. A smooth muscle layer is not incorporated into our system and therefore our in vitro bleeding model includes neither smooth muscle vasoconstriction nor the effects of subendothelial tissue damage, such as smooth muscle cell-derived TF, which may, in turn, underestimate the role of fibrin. Although we collected data with TF-coated vascularized channels, TF on the microchannel surface may not biologically function as identically as TF from smooth muscle cells. Despite these limitations, because our model more accurately recapitulates the microvasculature where smooth muscle cells are less prevalent, especially on the venular and venous side, and TF may have less of a role[7], our data are likely not significantly affected. Indeed, we believe that this system is best leveraged as a complementary technology when used in conjunction with in vivo bleeding models as the limitations of one system can be addressed by the other. For example, murine bleeding models such as the cremaster laser injury model obviously involve all physiologic aspects of hemostasis[35, 36]. However, these experiments are relatively costly and involve significant amounts of labor while the mechanism of injury may not be physiologic, which may confound the data collected and possibly lead to artefact. The vascularized microfluidic bleeding model presented here recapitulates a true mechanical injury and enables the use of human (and therefore patient) blood samples, while the experiments themselves are relatively inexpensive and simple to conduct.

With the capability to integrate an intact endothelium, physiologic flow conditions, a controlled mechanical injury, and human whole blood in a controlled manner, our vascularized microfluidic bleeding model is ideal for studying the global hemostatic potential of patient samples in a microvascular model and for dissecting pathophysiologic mechanisms of disease. In addition, with the capability to holistically assess and visualize the entire hemostatic process, this system can immediately aid the fields of experimental hematology and vascular biology as a novel research enabling tool. Moreover, these attributes also enable the system to potentially function as an entirely new category of diagnostic for bleeding disorders, although the standardization and characterization of the in vitro endothelium is needed and will be the basis of upcoming studies. Finally, this endothelialized microfluidic-based in vitro bleeding model also holds significant promise as a drug discovery platform, potential diagnostic, and therapy-guiding tool for hemostatic and thrombotic disorders.

## Methods

**Sample collection.** Human blood was collected according to our institutes' Institutional Review Board-approved protocols per the Declaration of Helsinki and conditioned for each experiment using the dyes, antibodies, and drugs as described as below in Methods and also in Results sections.

**Reagents and antibodies.** Plasma membrane stain (CellMask Deep Red, C10046), fluorescent-tagged fibrinogen (F13191), annexin V (A13202), fluorescently tagged secondary antibodies against mouse IgG (A11001, A11004, 1:400), and calcein (C34852, 2 μg ml⁻¹) were purchased from Life Technologies. Eptifibatide (Integrilin) was purchased from COR Therapeutics. Antibodies were purchased from Fisher Scientific (CD41, NB100-2614, 1:400), Santa Cruz Biotechnology (P-selectin, sc-19996, 1 μg ml⁻¹), Bio-Rad (vWF-FITC, AHP062F, 1:1000), BD (CD45, 555483, 1:400), and abcam (TF, ab35807, 1:200). Antibody AVW-3 and 59D8 (Alexa Fluor 488 conjugated) were provided by Dr. Shawn M. Jobe (Blood Center of Wisconsin, 10 μg ml⁻¹, 1:100) and mAb 2–76 by Dr. Shannon L. Meeks (Emory University, 10 μg ml⁻¹)[37, 38].

**Microfluidic device construction.** The microfluidic devices used in this study were fabricated using soft lithography with PDMS (Ellsworth Adhesive) in three separate layers: vascular layer, valve layer, and valve actuator. The vascular layer (with vascular channel 150 μm width and 50 μm height) was oxygen plasma bonded using a plasma cleaner (Harrick Plasma, PDC-32G) to the valve layer (50 μm thick), which contained a partially shielded area to prevent oxygen plasma treatment and, in turn, to create the movable pneumatic valve (Supplementary Fig. 7). Inlet and outlet holes were then punched into the bonded layers and oxygen plasma bonded to the valve actuator (with valve channel, 6 μm in height).

**Endothelialization of the microfluidic device.** All the microchannels (vascular, outlet, and valve) of pre-vacuumed devices were infused with collagen type 1 (BD, 1 mg ml⁻¹), and then only the vascular channel was infused with fibronectin (Sigma-Aldrich, 50 μg ml⁻¹) before HUVECs or HAECs (Lonza) were seeded at a concentration of 10 million cells ml⁻¹ in 8% Dextran (Sigma-Aldrich) in their growth media (EGM-2, Lonza). Fifteen minutes after cell adhesion, the device was washed with fresh media and further incubated at 37 °C for 2 h before the syringe filled with cell growth media was connected and media was perfused into the vascular channel using a syringe pump (PhD Ultra, Harvard Apparatus) at shear rates of 500 or 2500 s⁻¹ until cells grew to confluence.

**Mechanical induction of vascular injury and bleeding assay.** Vascular injury was introduced by making an opening between vascular layer and valve layer, which were not covalently bonded together, by inducing positive hydraulic pressure with a syringe, while pulling the valve layer downward using a separate syringe pump. Outlet tubing in the outlet channel was then cut at the base of the device to achieve ambient pressure. Whole blood collected in sodium citrate buffer (BD) and treated with CTI (40 μg ml⁻¹, Haematologic Technologies) was conditioned and recalcified before the experiments and perfused into the channel using a syringe pump at shear rates of 500 or 2500 s⁻¹. Time lapse images were collected approximately one image every 10 s (6 frames per minute) by a confocal microscope (LSM 700, Zeiss) during the experiments and bleeding time was measured.

**Data analysis.** Statistical analysis (Mann–Whitney test and one-way analysis of variance) of bleeding time was performed and graphed with Graphpad Prism 5. Fluorescence intensity, co-localization (Pearson correlation coefficient), and area

were measured by Image J (NIH) and Zen (Zeiss) and statistically analyzed by Student's *t*-test. *P*-value < 0.05 was considered as significant. COMSOL was used for computational fluid dynamics simulations. The intensity map of fluorescent signals was created with MATLAB (MathWorks).

**Data availability**. The datasets generated and/or analyzed during the current study are available from the corresponding author on reasonable request.

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

## Acknowledgements
Financial support was provided by NIH R01 (HL121264, HL130918), NIH U54 (HL112309), and NSF CAREER (1150235) to W.A.L. Also, this work was performed in part at the Georgia Tech Institute for Electronics and Nanotechnology, a member of the National Nanotechnology Coordinated Infrastructure, which is supported by NSF ECCS (1542174).

## Author contributions
W.A.L., S.M.J., B.A., Y.S., D.R.M., and Y.Q. designed the study. B.A. designed the device. E.T.H. constructed the devices. Y.S. and B.A. conducted experiments. R.T., M.E.F., and R.G.M. performed computer simulation and heatmap creation. Y.S. and M.F. did data analysis with help by R.G.M., R.T., J.C.C., and M.A.C. W.H.B., S.L.M., and S.M.J. provided antibodies. Y.S., W.A.L., E.T.H., B.A., R.T., J.C.C., D.R.M., M.A.C., G.E.G., and S.M.J. wrote the manuscript.

## Additional information

**Competing interests:** The authors declare no competing financial interests.

