## [Peer Review File · Nature Communications]

Reviewers' comments:

Reviewer #1 (Remarks to the Author):

The authors of the manuscript titled "A Microengineered, Vascularized Bleeding Model that Integrates the Principal Components of Hemostasis" develop a novel microfluidics device that mimics the process of hemostasis. The authors describe a microfluidic device with endothelium on its walls and pneumatic valve to initiate real-time bleeding. The current models with microfluidic devices that enable hemostasis are limited therefore this work is interesting to the hemostasis community. I have the following comments:

Device fabrication:

The author didn't mention what is coated on the wound/outlet channel.

The fabrication details and the mechanical functionality of the device are not clear. Does the valve only two positions on/off or does it have any intermediate controllable positions? If the valve has just on/off positions - the wound length would be just the width of the channel. This creates a substantial change in the hemodynamics that needs to be addressed.

Results section:

In the 47 experiments, the lengths of injury were different and the mean length was 132.49 ± 40.2 microns. The authors conclude that bleeding time doesn't depend on initial wound size or shear stress - through the Fig 2 in supplementary. Platelets immediately adhere at the wound site and reduce wound size, - the result seems very interesting but the reason is not concrete enough, as longer injury should take longer to seal if flow rate and everything else is maintained.

Platelet adhesion, fibrin accumulation and bleeding time were all same under arterial and venous shear conditions. This is another interesting result, which is not reasoned by the authors. Previous studies with collagen/TF have shown the difference in thrombosis in arterial and venous shear rates. It is not clear how hemostasis wasn't achieved in the absence of endothelial cells. How did the authors identify the expression of P selectin, did they use any chemical compound that highlights it?

vWF inhibition on endothelial cells is achieved through the use of AVW3 protein. This protein affects only vWF platelet binding without affecting other functions as stated by the author. By inhibiting vWF observations proved that at high shear rates no hemostasis was achieved while at low shear rates nothing changed from normal conditions. It would be better if these results are validated with any animal model or in vivo data.

PSer is a lipid found in the cell membrane. The authors say that platelets in the experiments were PSer negative, but fail to mention why they are PSer or how they were made PSer negative. Based on this fact, they prove that PSer in the injured endothelial cells serves as a prothrombotic surface. In fact, patients with platelet PSer defects, are said to have Scott syndrome which is a mild bleeding disorder. It would be interesting to see how PSer negative platelets responded to hemostasis, in terms of bleeding time.

By inhibiting factor VIII, they proved that hemophilic blood will result in unstable clot leading to re-bleeding continuously. Though these findings are already understood in the medical community, this device re-establishes the fact and opens doors for use in diagnostics.

There are recently published papers in *CAMB* "A Microfluidic Model of Hemostasis Sensitive to Platelet Function and Coagulation" also develops a platform to measure hemostasis and a subsequent paper in the same journal discusses the spatial and temporal dynamics of thrombosis. These seem relevant to this work.

Figures:

Fig. 2 has bleeding times for many samples plotted, and I can see a significant difference in bleeding time from 250 s to 1000 s. Why is there such a difference in bleeding times even among healthy donors? Is there an explanation for a similar plot in figure 3 and 5?

Reviewer #2 (Remarks to the Author):

This paper describes a new microfluidic model for hemostasis, including induced injury of endothelial cell-lined channels, and study of the role of platelets, leukocytes, fibrin, phosphatidyl serine, and von Willebrand factor. Although most of the basic findings here were already well known from a variety of other studies, there were still some surprises. Despite the fact that there have been other microfluidic models of hemostasis and thrombosis, most have focused on specific aspects of bleeding or thrombosis and none have been as complete. This model has the potential to be much more informative than the typical assays used to study or diagnose disorders of hemostasis.

Comments and critique:

1. Probably the most used models of hemostasis and thrombosis beyond the very limited assays used in clinical coagulation laboratories are various mouse models. Some of these authors know very well the limitations of mouse models, since in other papers they have characterized the most commonly used one, the ferric chloride induced vascular injury. It would be worthwhile to include in the Discussion of this paper a very brief summary of the strengths and limitations of these mouse models, especially the cremaster laser injury model, in comparison with this microfluidic model.
2. It would also be worthwhile to emphasize more strongly the advantage of this model in terms of using human cells versus mouse models, where there are species differences in aspects of hemostasis.
3. Since this model has single cell resolution, as pointed out in the manuscript, why do the authors characterize clot contraction in terms of saturated fluorescence, rather than visualizing individual platelets becoming more closely packed?
4. Eptifibatide inhibits platelet aggregation. It is not surprising that there would be the same density of platelets initially adhering to the injured surface, but I would expect more platelets to aggregate on top of those platelets in the control than with eptifibatide. The authors should describe the difference in the total number of platelets at the wound site.
5. Why was the valve size 6 μm ? Were other sizes tested in the initial design of the system?
6. Can the authors discuss the relationship between injury size and extent of endothelial damage?
7. The P_{Ser} exposure on endothelial cells being more significant than platelet P_{Ser} has been reported, but I think that the timing of these events is new information and should be emphasized.
8. In addition, references are needed for the mention of the controversy over P_{Ser} exposure on endothelial cells versus platelets.
9. The effects of TF on leukocytes is quite interesting. However, TF on the surface of the channel is not the same as TF from smooth muscle cells. This should be emphasized.
10. The results of the hemophilia experiments are interesting. Can the authors quantify in any way the instability of these clots? Perhaps by measuring fluorescent material coming off the clot downstream of the injury?
11. The authors measure bleeding by escaping RBCs, which is fine. However, there can be plasma leakage well beyond the cessation of RBC escape, as shown by the mouse cremaster injury model. Can the authors measure and quantify plasma leakage, perhaps by including a small fluorescently labeled marker in the plasma?

12. From the videos, there is extensive activity (platelets, fibrin, etc) in the "extravascular space." Can this be analyzed?

13. What is the image collection rate? This should be mentioned in terms of the rates of movement of the cells, especially in the arterial model.

14. What is the diagnostic potential of this model, since it is certainly much more powerful than current clinical coagulation tests? This should be mentioned.

15. The Discussion mentions differences for larger vessels in terms of Reynolds, Dahnkohler and Peclet numbers. For non-engineers reading this paper, this should be rephrased or described.

Reviewer #3 (Remarks to the Author):

General:

This study describes the design and testing of a microfluidics device that includes an endothelialized channel and a novel means for causing a localized injury without opening or penetrating the device. This goal is a worthy one: in contrast to most other devices that have been published, the device described here is intended to recapitulate a hemostatic injury in the microvasculature. However, reading the manuscript required considerable patience since, as noted below, key details were missing in the text and figure legends, making it hard to be sure which conditions were present in different assays. This issue aside, my principal reservation about the device design is that although "bleeding" stops, most of the observations are made in the absence of tissue factor (TF) and are shown to be TF-independent. As a result, achieving hemostasis appear to be more dependent on the intrinsic pathway of coagulation and, possibly, endothelial cell PS exposure than is likely to be the case in vivo. Therefore, even though the authors have highlighted their ability to detect effects of VWF blockade and hemophilia A, it is not clear that this means that their results necessarily reflect what occurs in vivo when injuries occur to a blood vessel wall that includes TF in at least the adventitia.

Specific issues:

1. Perhaps to keep down the length of the manuscript, critical information is missing. For example, I didn't see the concentration of CTI used in the blood that was perfused into the chamber. I also couldn't find legends for the videos and frequently had to hunt to figure out which color represented what in the fluorescence images.

2. Line 42 and lines 205-211: "the importance of endothelial PS in hemostasis" - The studies show that in this device, endothelial cells near the wound site become annexin V positive. I'm not sure that this observation necessarily shows that the endothelial cells are promoting thrombin production, nor does it show that this happens in vivo.

3. line 95: "eptifibatide predominantly affects hemostatic plug formation via inhibition of platelet contraction, and therefore clot density." Although this is stated in the text, it is not apparent in the figures provided.

4. lines 171-175: "contraction of these aggregates was more pronounced in control blood than in eptifibatide-treated blood". This isn't apparent in the video. Quantification and an analysis of multiple injuries is needed to allow readers to draw their own conclusions.

5. lines 176-179: "Higher magnification visualization of the bleeding site revealed clear co-localization of fibrinogen/fibrin with platelets in control conditions but not in the eptifibatide-

treated condition (Fig. 2e). Overall, these results indicate that the integrin $\alpha\text{IIb}\beta\text{3}$ -fibrin interaction “glues” platelets together at the site of vascular injury, but that this interaction alone is not necessary for hemostasis in the microvasculature.” This isn’t apparent in the figure. In addition, co-localization of the 2 fluorophores does not necessarily imply binding of platelets to fibrin(ogen), and using fluorescent fibrinogen limits the ability to conclude which is fibrin and which is fibrinogen. Finally, quantification and an analysis of multiple injuries should be included to allow readers to draw their own conclusions.

6. lines 190-194 and figure 3: I may have missed it, but the text doesn’t readily indicate the difference between Figure 3a and 3b. It also isn’t stated why there are so few replicates of everything except the control? Would the conclusions hold up if there were more data?

7. lines 220-225: “To assess the effect of exogenous TF in our system, vascular microchannels were pre-coated with 1 nM of TF and collagen prior to the seeding endothelial cells. We observed enhanced fibrinogen/fibrin accumulation (Supplementary Fig. 5b) at early time points with only a slight reduction in the bleeding time. These preliminary results highlight the utility of our system in assessing the synergistic relationship between coagulation factors in the hemostatic process.” This experiment is especially important. The hope for this model is that it will recapitulate in vivo events. Although the role of the intrinsic It would be a good idea to quantify the data on fibrin(ogen) accumulation and the bleeding time, with sufficient replicates.

We deeply appreciate the time and thoughtful consideration that the editor(s) and reviewers have given to our manuscript. Overall, we are extremely pleased that all 3 reviewers find our work of interest. Specifically, Reviewer 1 recognized the innovation of our system in that “current models with microfluidic devices that enable hemostasis are limited therefore this work is interesting to the hemostasis community.” Moreover, Reviewer 2 echoed this sentiment, stating that while “other microfluidic models of hemostasis and thrombosis...focused on specific aspects of bleeding or thrombosis, none have been as complete” as our system and that our technology “has the potential to be much more informative than the typical assays used to study or diagnose disorders of hemostasis.” Finally, Reviewer 3 noted that the goal of our work “is a worthy one” and that “in contrast to most other devices that have been published, the device described here is intended to recapitulate a hemostatic injury in the microvasculature.” As requested by the reviewers and editor, we believed we have addressed the reviewers’ concerns and have made significant changes to our manuscript based on their feedback and suggestions. Specifically, we have:

1. Added 1 new and 2 modified supplementary figures and 3 new supplementary videos.
2. Substantially rewritten and added additional explanation to the experimental methods, results, and discussion, and specifically added a new supplementary document where all the supplementary videos are explained in detail
3. We added new subfigures (panels) in Figure 2 and 5.

Reviewers' comments:

Reviewer #1 (Remarks to the Author):

The authors of the manuscript titled “A Microengineered, Vascularized Bleeding Model that Integrates the Principal Components of Hemostasis” develop a novel microfluidics device that mimics the process of hemostasis. The authors describe a microfluidic device with endothelium on its walls and pneumatic valve to initiate real-time bleeding. The current models with microfluidic devices that enable hemostasis are limited therefore this work is interesting to the hemostasis community.

We thank the reviewer for recognizing the innovation of our system, how it is differentiated from “current models with microfluidic devices” and how it is of interesting to the scientific community interested in hemostasis.

I have the following comments:

Device fabrication:

The author didn't mention what is coated on the wound/outlet channel.

We apologize for the lack of clarity on what the wound/outlet channel is coated with, it is pre-filled with collagen type I (1 mg /ml). We added this to the method section as below in line 378-380; (**Bold** is new text);

All microchannels (vascular, outlet, and valve) of pre-vacuumed devices were infused with collagen type 1 (BD, 1 mg/ml) and then **only the vascular channel was infused** with fibronectin (Sigma-Aldrich, 50 µg/ml)

The fabrication details and the mechanical functionality of the device are not clear. Does the valve only two positions on/off or does it have any intermediate controllable positions? If the valve has just on/off positions - the wound length would be just the width of the channel. This creates a substantial change in the hemodynamics that needs to be addressed.

We apologize for the lack of clarity on this issue and have put forth our best effort to address this issue in our manuscript. The valve membrane has two positions - closed or open - and for our studies, the valve membrane deflects to the open position, which corresponds to a 6 µm height opening. The reviewer is correct, for a given device (which has a specific valve height), the height of the wound is constant, and therefore channel width is the only factor to mediate the wound length. To address the Reviewer's point, we characterized the hemodynamics of this process. Specifically, COMSOL simulations demonstrate different shear stress profiles with different channel widths (all of which we have observed experimentally to be attainable within the range of the device's capabilities). However, within this dynamic range, the shear stress profiles were all within the same order of magnitude (see the figure below, which we have now also included as new Supplementary Figure 2b).

Therefore, the variability of channel widths/wound lengths do not significantly affect the *in vitro* bleeding time of our system, as evidenced by the measured bleeding times obtained from a large number of experiments (n=47) (Supplementary Figure 2a). In addition, in light of the new findings, we have altered the relevant text in the Results section (line 153-157) to now read (new text is in **bold**):

We observed that the initial wound width and bleeding time did not correlate in our studies (Supplementary Fig. 2a, $R^2= 0.088$), **which is likely due to the fact that the shear stress profiles are all within the same order of magnitude over the operable range of wound widths in our system**

(Supplementary Fig. 2b). Therefore, in our system, the time course of hemostasis is not dependent on initial wound width.

However, altering the height of the valve actuator does affect the bleeding time, as the area of wound opening is altered dramatically. We have since fabricated and tested different devices with valve actuators of different heights to determine and characterize the effect this parameter has on the “*in vitro*” bleeding time in our microdevices (please see our response to the next comment below for more details).

Results section:

In the 47 experiments, the lengths of injury were different and the mean length was 132.49 +/-40.2 microns. The authors conclude that bleeding time doesn't depend on initial wound size or shear stress – through the Fig 2 in supplementary. Platelets immediately adhere at the wound site and reduce wound size, - the result seems very interesting but the reason is not concrete enough, as longer injury should take longer to seal if flow rate and everything else is maintained.

We thank the reviewer for the opportunity for us to clarify this issue. Our more recent results demonstrate that wound size does affect the *in vitro* bleeding time but the dominant parameter is not the width of the wound (as the shear stress profiles are all within the same order of magnitude over the operable range of wound widths in our system as stated above) but the height of the wound, which is dictated by the height of the valve in its open position. Our most commonly used device in this manuscript has a valve height of 6 μm . We have developed and tested other devices with different valve heights of 2.7, 22, and 50 μm . Using devices with 2.7 μm wound heights significantly shortened the *in vitro* bleeding time. However, with 2.7 μm wound heights, the height of the opening is too low for blood cells such as RBCs and WBCs (with 6-10 μm diameters) to traverse and the hemodynamics become significantly more complicated and involve pathological flow rates/shear stresses. In devices with wound heights of 50 μm , the majority of the blood flowed into the wound channel (as expected) rather than the vascular channel and hemostasis was not achieved within the 20 minute standard time period used in this study, thus 50 μm wound height is not suitable. Devices with wound heights of 22 μm also had significantly increased bleeding time (median value 1203 sec) compared to devices with wound heights of 6 μm . However, with prolonged recording time, blood samples in the syringe pump were able to separate which could confound our data. Ultimately, we concluded that the devices with 6 μm wound heights enabled the widest dynamic range with bleeding times that were physiologically relevant that also yielded the more precise control of the bleeding and hemostatic process (i.e. no concerns about sample separation upstream). While this latter issue can certainly be addressed by engineering an apparatus that continuously mixes the sample so that the blood sample does not separate, we feel it is beyond the scope of this manuscript.

Platelet adhesion, fibrin accumulation and bleeding time were all same under arterial and venous shear conditions. This is another interesting result, which is not reasoned by the authors.

The reviewer brings up an interesting point and we apologize for the lack of clarity. We do in fact observe that the bleeding time remains unchanged under arterial and venous shear conditions with HAECs and HUVECs. We conjecture that under those different condition our system, in which there is a mechanical injury of relatively constant size that is induced at a specific site of an endothelialized channel, two opposing phenomena are occurring. First, the increased pressure and velocity under arterial conditions should lead to more “bleeding” in our system. Second, the increased flux of hemostatic components including platelets and coagulation factors should lead to a more rapid formation of a hemostatic plug, especially given the fact that the injury size is unchanged, which then diverts blood flow back to the vascular channel and additional platelet/fibrin accumulation ceases, which is what occurs physiologically. In addition, the endothelial cells themselves, which are from venous and arterial vascular beds, may also be biologically more “tuned” to their corresponding shear conditions. These phenomena therefore likely collectively cancel each other out leading to similar hemostatic plug formation and bleeding times under arterial and venous conditions. We have added these points to the Discussion in the revised manuscript (Line 317-330). More systemic studies need to be conducted to prove this hypothesis with our system and are currently underway but we believe are beyond the scope of this manuscript.

Previous studies with collagen/TF have shown the difference in thrombosis in arterial and venous shear rates. It is not clear how hemostasis wasn't achieved in the absence of endothelial cells.

We apologize for the lack of clarity on this issue. In our system, hemostasis was not achieved in the absence of endothelial cells as the vascular microchannels are coated with collagen I and fibronectin, even we observed diffuse platelet adhesion with some fibrin(ogen) accumulation in both vascular and wound channels. This is consistent with the recent results described by the Neeves group in which they demonstrated that the combination of collagen/TF (but not collagen only) is required to achieve full closure of a microchannel opening using whole blood (Schoeman ,et al., Cell Mol Bioeng, 2017, Ref#17) . This suggests that TF or prothrombotic activity from endothelial cell damage are important for complete hemostasis to be achieved in our system.

How did the authors identify the expression of P selectin, did they use any chemical compound that highlights it?

Expression of P-selectin was assessed via the use of a P-selectin antibody. We apologize for the confusion on this issue, which has been clarified in our Results section (Line 168).

vWF inhibition on endothelial cells is achieved through the use of AVW3 protein. This protein affects only vWF platelet binding without affecting other functions as stated by the author. By inhibiting vWF observations proved that at high shear rates no hemostasis was achieved while at low shear rates nothing changed from normal conditions. It would be better if these results are validated with any animal model or in vivo data.

We apologize for the confusion regarding the nature of AVW3. AVW3 is an antibody that has been historically developed and characterized (Hillery, et al. Ref#37) and has a high specificity to the platelet GPIb binding motif within the A1 domain of VWF. We agree that an animal model of this would be ideal and in fact the Blood Center of Wisconsin currently has a mouse model that is still being characterized with a 2B phenotype to be used to further our studies.

PSer is a lipid found in the cell membrane. The authors say that platelets in the experiments were PSer negative, but fail to mention why they are PSer or how they were made PSer negative. Based on this fact, they prove that PSer in the injured endothelial cells serves as a prothrombotic surface. In fact, patients with platelet PSer defects, are said to have Scott syndrome which is a mild bleeding disorder. It would be interesting to see how PSer negative platelets responded to hemostasis, in terms of bleeding time.

We apologize for the confusion here. The platelets we used were obtained from healthy donors and in our manuscript, we denote "PSer positive" and "PSer negative" platelets to mean platelets that expressed or did not express PSer on the platelet surface, respectively. PSer positivity was determined with the Annexin V stain, which labels surface bound PSer. In our manuscript, "PSer negative" platelets were normal platelets that did not stain positively for Annexin V as the PSer was bound to the inner leaflet of the platelet membrane and therefore not accessible to the Annexin V label that are external to the platelets. As such, platelets described as "PSer negative", because they are from normal subjects, do in fact still have PSer, but the PSer is not expressed at the cell surface. We have clarified this point in the manuscript at line213-217. Moreover, the purpose of this experiment was to determine whether endothelial PSer or platelet PSer was the dominant source of PSer exposure at the site of vascular injury and at what timepoints. We agree completely with the reviewer that conducting experiments with Scott syndrome platelets would be extremely interesting and worthwhile, but are beyond the scope of this paper, especially given that we do not readily have access to those patients at our local clinical centers.

By inhibiting factor VIII, they proved that hemophilic blood will result in unstable clot leading to re-bleeding continuously. Though these findings are already understood in the medical community, this device re-establishes the fact and opens doors for use in diagnostics.

We thank the reviewer for this constructive comment and completely agree that this highlights the diagnostic potential of our system.

There are recently published papers in CAMB "A Microfluidic Model of Hemostasis Sensitive to Platelet Function and Coagulation" also develops a platform to measure hemostasis and a subsequent paper in the same journal discusses the spatial and temporal dynamics of thrombosis. These seem relevant to this work.

We thank the reviewer for this suggestion. We have cited and discussed the first paper mentioned above (Schoeman, et al. 2016, Ref#17) and added the second paper to the references (Zilberman-Rudenko, et al. 2016, Ref#X15).

Figures:

Fig. 2 has bleeding times for many samples plotted, and I can see a significant difference in bleeding time from 250 s to 1000 s. Why is there such a difference in bleeding times even among healthy donors? Is there an explanation for a similar plot in figure 3 and 5?

We thank the reviewer for this excellent question and apologize for not originally including an explanation related to the variance and range of the bleeding time. We believe that donor-to-donor variation causes the observed differences, similar to that seen in the historical template bleeding time assay. In this test, a trained technician used a template to make an incision on the volar forearm surface and subsequently monitored the time to hemostasis. As might be expected, the ability and the repeatability of the technician could affect the results, however, some technicians demonstrated exceptional repeatability as measured by repeat testing on individuals (Mielke, et al., 1969). When raw data was reported, the healthy subjects had recorded bleeding times ranging from 3 minutes to 12 minutes (Mielke, et al., 1969). In a similar study with a repeatable technician, we can extrapolating the range from 2 standard deviations around the mean to be 1.5 minutes to 7.5 minutes (Harker and Slinchter, 1972). While the historic bleeding time test encompasses a very global assessment of hemostasis that includes vasoconstriction and vascular tone, it is not unreasonable to conclude that some of the variation observed in the bleeding time occurs due to differences in blood composition and activation state between donors. As would be expected, our system is also sensitive to this phenomenon. Although the exact mechanistic reasons for this variation remain unknown, we are excited about the prospect of using our system to explore this phenomenon given its ability to precisely control the wound and hematological parameters.

In regards to the control and healthy in Figure 3 and 5 are same samples, perfused at 500 s⁻¹. We have many of this default normal condition we performed initially (n=47). We wanted to be thorough, and

have included all the experimental results in Figure 3 and 5.

Reviewer #2 (Remarks to the Author):

This paper describes a new microfluidic model for hemostasis, including induced injury of endothelial cell-lined channels, and study of the role of platelets, leukocytes, fibrin, phosphatidyl serine, and von Willebrand factor. Although most of the basic findings here were already well known from a variety of other studies, there were still some surprises. Despite the fact that there have been other microfluidic models of hemostasis and thrombosis, most have focused on specific aspects of bleeding or thrombosis and none have been as complete. This model has the potential to be much more informative than the typical assays used to study or diagnose disorders of hemostasis.

We thank the reviewer for noting the significant potential of our system and noting the attributes of our system.

Comments and critique:

1. Probably the most used models of hemostasis and thrombosis beyond the very limited assays used in clinical coagulation laboratories are various mouse models. Some of these authors know very well the limitations of mouse models, since in other papers they have characterized the most commonly used one, the ferric chloride induced vascular injury. It would be worthwhile to include in the Discussion of this paper a very brief summary of the strengths and limitations of these mouse models, especially the cremaster laser injury model, in comparison with this microfluidic model.

We thank the reviewer for this insightful point and accordingly, we have added more text in the Discussion describing our system in the context of the mouse injury models. We believe that our device is best used as a complementary technology in conjunction with *in vivo* bleeding models as the disadvantages of one system can be addressed by the other.

To that end, we have added the following to the manuscript, line 335-347. **Bold** is new text

because our model more accurately recapitulates the microvasculature where smooth muscle cells are less prevalent, especially on the venular and venous side, our data is likely not significantly affected. **Indeed, we believe that this system is best leveraged as a complementary technology when used in conjunction with *in vivo* bleeding models as the limitations of one system can be addressed by the other. For example, murine bleeding models such as the cremaster laser injury model obviously involve all physiologic aspects of hemostasis. However, these experiments are relatively costly and**

involve significant amounts of labor while the mechanism of injury may not be physiologic, which may confound the data collected and possibly lead to artifact. The vascularized microfluidic bleeding model presented here, recapitulates a true mechanical injury and enables the use of human (and therefore patient) blood samples, while the experiments themselves are relatively inexpensive and simple to conduct.

2. It would also be worthwhile to emphasize more strongly the advantage of this model in terms of using human cells versus mouse models, where there are species differences in aspects of hemostasis.

The reviewer makes an excellent point and we have integrated the capability of using human cells into the Discussion as described above.

3. Since this model has single cell resolution, as pointed out in the manuscript, why do the authors characterize clot contraction in terms of saturated fluorescence, rather than visualizing individual platelets becoming more closely packed?

The reviewer makes an excellent suggestion and to that end, we conducted new experiments to address that specific point. Individual platelets labeled with the same color would be difficult to resolve optically due to the high number of cells and the fact that they aggregate. Therefore, in our new experiments, 1-2% of platelets were stained with a dye with different fluorescent (calcein, green), so individual platelets could be tracked as they integrate into the nascent clot and then over the time course of hemostasis. This direct observation proves that we can observe contraction of platelet aggregates at the single cell level in our system.

The results of these experiments are represented in the new Supplemental videos 3 and 4 and specifically stated in the Results section line 179-182. In Supplemental video 3, healthy control blood is perfused and a subpopulation of platelets (stained in green) integrate into the nascent and as the clot contracts, those platelets are visibly displaced proximally and against the direction of flow in the vascular channel during the bleeding. In Supplemental video 4, eptifibatide-treated blood is used and labeled platelets do not displace from the position where they originally integrate into the nascent clot, which demonstrates that clot contraction is significantly attenuated in this case.

4. Eptifibatide inhibits platelet aggregation. It is not surprising that there would be the same density of platelets initially adhering to the injured surface, but I would expect more platelets to aggregate on top

of those platelets in the control than with eptifibatide. The authors should describe the difference in the total number of platelets at the wound site.

We thank the reviewer for this suggestion. In Figure 2b, we compare the total area of adherent platelets in the control and eptifibatide-treated conditions, which did not demonstrate a significant difference. In addition, we analyzed the total fluorescence intensity of the adherent platelets, which correlates with the total number of the platelets at the wound site. As shown below, we did not observe any significant difference between the two conditions. However, this analysis does have a limitation in that it is not a 3D analysis and there may be parts of the platelet aggregate in which fluorescence intensity was saturated and therefore underestimate the number of platelets. However, we believe that the numbers of the platelets recruited at the wound site would not be significantly different with 3D analysis.

5. Why was the valve size 6 μm ? Were other sizes tested in the initial design of the system?

The reviewer brings up an excellent point, and we originally developed devices with different valve heights of 2.7, 6, and 50 μm . Our most commonly used device has a valve height of 6 μm , which we find is the optimal device in terms of dynamic range and physiologic relevance and therefore is the valve height in which all of our experiments was conducted in this manuscript. Using devices with 2.7 μm wound heights significantly shortened the *in vitro* bleeding time (median value was 289 sec compared to 409 sec with devices with wound heights of 6 μm as below). However, with 2.7 μm wound heights, the height of the opening is too low for blood cells such as RBCs and WBCs (with diameters of 6-10 μm) to

traverse and the hemodynamics become significantly more complicated and involve pathological flow rates/shear stresses. In devices with wound heights of 50 μm , the majority of the blood flowed into the wound channel (as expected) rather than the vascular channel and hemostasis was not achieved within the 20minutes standard time period used in the study, thus 50 μm wound height is not suitable for this study. Devices with wound heights of 22 μm also significantly increased bleeding time (median value 1203 sec) compared to devices with wound heights of 6 μm . However, with prolonged recording time, blood samples in the syringe pump could separate which could confound our data. Ultimately, we concluded that the devices with 6 μm wound heights enabled the widest dynamic range with bleeding times that were physiologically relevant that also yielded the more precise control of the bleeding and hemostatic process (i.e. no concerns about sample separation upstream). While this latter issue can certainly be addressed by engineering an apparatus that continuously mixes the sample so that the blood sample does not separate, we feel it is beyond the scope of this manuscript. The data below shows the bleeding time of different wound heights.

6. Can the authors discuss the relationship between injury size and extent of endothelial damage?

When the wound is mechanically induced, a few endothelial cells (typically just one) are near-instantaneously damaged and detached regardless of the wound widths that were within the range of our system. This damaged area causes endothelial cell detachment and exposes the subendothelial matrix. We measured those damaged/exposed areas using healthy control blood (n=18 experiments) and found little correlation between the damaged/exposed areas and wound width as well as bleeding time (please see plots below). This may imply that, in our range of wound widths, the various factors involved with hemostasis (coagulation factors, platelets, biological function of the neighboring

endothelial cells that are still detached) interact and compensate for each other to achieve hemostasis at similar time scales.

7. The PSer exposure on endothelial cells being more significant than platelet PSer has been reported, but I think that the timing of these events is new information and should be emphasized.

We thank the reviewer for this excellent suggestion.

To that end, we have added the following to the manuscript, line 304-309. **Bold** is new text

In our system, we clearly demonstrate that PSer is exposed on injured endothelial cells within seconds of a mechanical vascular injury and expression continues to increase in the direct vicinity of the injury over the time course of hemostasis while platelet PSer expression at the wound site occurs later. **While PSer exposure on endothelial cells has been reported as being more significant than platelet PSer, the timing of these different events has been unresolved and debated.**

8. In addition, references are needed for the mention of the controversy over PSer exposure on endothelial cells versus platelets.

Per the reviewer's request, we have added these references below that cite the controversy over PSer exposure on endothelial cells versus platelets.

Ivanciu, et al. (2014, ref#18), Bevers, et al (1982, ref#29), Bouchard, et al. (2001, ref#30), Monroe, et al. 2002, ref#31), Heemskerk, et al. (2013, ref#32), Abaeva, et al. (2013, ref#33), Fujii, et al. (2015, ref#34)

9. *The effects of TF on leukocytes is quite interesting. However, TF on the surface of the channel is not the same as TF from smooth muscle cells. This should be emphasized.*

The reviewer again makes an excellent point and we have modified the text to emphasize this.

To that end, we have added the following to the manuscript, line 330-335. **Bold** is new text

A smooth muscle layer is not incorporated into our system and therefore our *in vitro* bleeding model includes neither smooth muscle vasoconstriction nor the effects of subendothelial tissue damage, **such as smooth muscle cell-derived TF. Although we collected data with TF-coated vascularized channels, TF on the microchannel surface may not biologically function as identically as TF from smooth muscle cells. Despite of these limitations**

10. *The results of the hemophilia experiments are interesting. Can the authors quantify in any way the instability of these clots? Perhaps by measuring fluorescent material coming off the clot downstream of the injury?*

This is an interesting question the reviewer poses. Regarding clot instability, we do not observe “embolic” phenomenon the reviewer asks about. However, in certain conditions we do see re-bleeding after the formation of a temporary hemostatic plug. This phenomenon occurs predominantly in the condition in which we model hemophilia A with inhibitor formation using an antibody against the A2 domain of Factor VIII (anti-A2 mAb 2-76). In those instances, re-bleeding occurs after an initial platelet plug is formed due to clot instability. The figure below shows a representative time plot of re-bleeding. **This new data are now included as a new panel in Figure 5 as the new Figure 5c.**

In addition, to address the reviewer’s question, we also measured the fluorescence profile of accumulating platelets across the wound area (i.e. line scans as function of distance from the valve/wound opening) to visually investigate how “porous” the clots are, as well as the fluorescence intensity which measures the platelet density across the wound as shown in the figure below. We did not observe any significant difference between healthy donor and hemophilia A patient blood samples. However, as shown in Figure 5f, there was little fibrin(ogen) deposition/formation at the wound site in blood from hemophilia A patients. Therefore, taken together, we believe these collective data suggests that the instability of the clot, and thus the lack of capability to achieve hemostasis, in blood from hemophilia A patient is due to lack of fibrin(ogen) to effectively fill and “glue” the clot together.

11. The authors measure bleeding by escaping RBCs, which is fine. However, there can be plasma leakage well beyond the cessation of RBC escape, as shown by the mouse cremaster injury model. Can the authors measure and quantify plasma leakage, perhaps by including a small fluorescently labeled marker in the plasma?

Again, the reviewer makes an interesting and good point regarding the capability to quantify plasma leakage in our system. First, we added the reference on cremaster injury model in the manuscript (Welsh, et al. 2013 and Stalker, et al. 2016) In addition, we conducted an experiment adding fluorescently-tagged bovine serum albumin as fluorescent molecular marker to visualize plasma leakage in our system. We did not detect any additional fluorescence leak into the extravascular space after the hemostatic plug was formed, as shown in the figure below. Therefore, we do not detect any obviously plasma leakage after the hemostatic plug is formed. While we must acknowledge that our microscopy system may not be sensitive enough to detect small amounts of plasma leakage, this most likely would not be a physiologically relevant amount.

12. From the videos, there is extensive activity (platelets, fibrin, etc) in the “extravascular space.” Can this be analyzed?

The reviewer makes an interesting and good point regarding the activities in the extravascular space in our system. We have collected the data on platelet and fibrinogen on this extravascular-wound channel, especially in the context of investigating if there are any difference between control and eptifibatide-treated blood. We observed the large variances even within the same conditions and therefore our results were not statistically different (please see the graphs on the next page).

One interesting observation we noted in the “extravascular space” during our experiments involves neutrophil crawling. This is easily visualized by phase contrast and fluorescence (via CD15 binding, please see new Supplementary video 7) microscopy. We compared the number of crawling neutrophils between vehicle control and eptifibatide-treated, as shown in the figure on the next page. Interestingly, even though bleeding times are similar in both conditions as shown as Figure 2a, eptifibatide-treated blood is associated with more neutrophils in the extravascular space than in the control condition. This implies that clots may be more mechanically favorable (less dense) for neutrophils to enable adhesion or migration into the extravascular space or that neutrophils are more activated in eptifibatide-treated blood, although the mechanisms remain unclear at this point. All of these observations require further investigation in future studies and we appreciate the reviewer’s suggestion.

13. *What is the image collection rate? This should be mentioned in terms of the rates of movement of the cells, especially in the arterial model.*

We apologize for the confusion on this point – our image collection rate is approximately one image every 10 seconds in our experiments (6 frames per minute). We have emphasized this in our Methods section (line 392-393).

14. *What is the diagnostic potential of this model, since it is certainly much more powerful than current clinical coagulation tests? This should be mentioned.*

We thank the reviewer for noting the diagnostic potential of our system. Per the reviewer’s suggestion, we have added the following to the Discussion section in the manuscript, line 351-359. **Bold is new text**

In addition, with the capability to holistically assess and visualize the entire hemostatic process, this system **can immediately aid the fields of experimental hematology and vascular biology as a novel research enabling tool. Moreover, those attributes also enable the system to potentially function as an entirely new category of diagnostic for bleeding disorders, although the standardization and**

characterization of the *in vitro* endothelium is needed and will be the basis of upcoming studies.

Finally, this endothelialized microfluidic-based *in vitro* bleeding model also holds significant promise as a drug discovery platform and therapy-guiding tool for hemostatic and thrombotic disorders.

15. The Discussion mentions differences for larger vessels in terms of Reynolds, Dahnkohler and Peclet numbers. For non-engineers reading this paper, this should be rephrased or described.

We apologize for the confusion regarding these dimensionless parameters. We have rephrased this section to describe the differences between larger vessels in a way that is more appropriate for non-engineers reading the paper. Specifically, we have added the following to the manuscript, line 311-317.

Bold is new text

No *in vitro* device can fully recapitulate all *in vivo* conditions, and our system is not without limitations. Due to its small scale, our microsystem physically models the microvasculature rather than larger vessels, which are dictated by **different fluid and mass transport phenomena due to the larger characteristic lengths and flow rates observed in larger vessels. The microscale dimensions of our system results in steady and laminar flow, which enables shear stress to be** tightly controlled and modulated to mimic venular/small venous to arteriolar/small arterial conditions, thereby enabling the dynamic physiologic range of the microcirculation.

Reviewer #3 (Remarks to the Author):

General:

This study describes the design and testing of a microfluidics device that includes an endothelialized channel and a novel means for causing a localized injury without opening or penetrating the device. This goal is a worthy one: in contrast to most other devices that have been published, the device described here is intended to recapitulate a hemostatic injury in the microvasculature. However, reading the manuscript required considerable patience since, as noted below, key details were missing in the text and figure legends, making it hard to be sure which conditions were present in different assays. This issue aside, my principal reservation about the device design is that although “bleeding” stops, most of the observations are made in the absence of tissue factor (TF) and are shown to be TF-independent. As a result, achieving hemostasis appear to be more dependent on the intrinsic pathway of coagulation and, possibly, endothelial cell PS exposure than is likely to be the case in vivo. Therefore, even though the authors have highlighted their ability to detect effects of VWF blockade and hemophilia A, it is not clear

that this means that their results necessarily reflect what occurs in vivo when injuries occur to a blood vessel wall that includes TF in at least the adventitia.

We thank the reviewer for noting the novelty and the worthwhile goal of our system. We do, however, apologize for the lack of clarity in terms of the specific conditions used in each experiment. To address this, we have stated the specific conditions for each experiment, and specifically we have added a new supplementary document where all the details of the supplementary videos are explained.

Regarding the issues with our system being TF-independent, no *in vitro* model perfectly recapitulates all *in vivo* conditions and we acknowledge that our TF-independent findings are a limitation our system. However, *in vivo*, the source of TF is predominantly from smooth muscle cells and our system more specifically models the microvasculature, where smooth muscle is less prevalent and therefore TF may have less of a role (Mackman, 2005, Ref#7). Furthermore, our device is also a microvascular mechanical disruption model that allows for evaluation of hemostasis components that are not dependent on smooth muscle or fibroblasts. As an example, the apparent endothelial contribution to hemostasis might be less evident in another model.

Based on the reviewer's comments, we have modified several points in the text accordingly. Specifically, we have added the following to the Discussion, lines 348-351. **Bold** is new text

With the capability to integrate an intact endothelium, physiologic flow conditions, a controlled mechanical injury, and human whole blood, in a controlled manner, our vascularized microfluidic bleeding model is ideal for studying the global hemostatic potential of patient samples **in a microvascular model** and dissecting pathophysiologic mechanisms of disease.

Specific issues:

1. Perhaps to keep down the length of the manuscript, critical information is missing. For example, I didn't see the concentration of CTI used in the blood that was perfused into the chamber. I also couldn't find legends for the videos and frequently had to hunt to figure out which color represented what in the fluorescence images.

We apologize for the lack of clarity, but the reviewer is correct in that we have been constrained with text limitations. That said, we have added a new supplementary document which lists the titles and legends for all videos that clarify all experimental conditions. Also please note that the CTI concentration is 40 µg/ml, which has also been clarified in the Methods section.

2. Line 42 and lines 205-2011: “the importance of endothelial PS in hemostasis” - The studies show that in this device, endothelial cells near the wound site become annexin V positive. I’m not sure that this observation necessarily shows that the endothelial cells are promoting thrombin production, nor does it show that this happens in vivo.

The reviewer makes an excellent point that we completely agree with. In our experiments reported in our manuscript, we do not show direct evidence that the PSer exposure by the damaged endothelial cells in our system generates thrombin. However, other groups have demonstrated that PSer exposed by endothelial cells do act a prothrombotic surface (Bombeli, et al. 1997 Ref#23). In addition, while our system does not provide direct evidence that the endothelial PSer generates thrombin, we do have evidence that fibrin formation co-localizes with the endothelial PSer. Specifically, we conducted an experiment where we perfused whole blood with a fluorescently-tagged antibody that is specific to fibrin (59D8-AF488) and not fibrinogen. As shown below, fibrin co-localizes at the edge of damaged endothelial cell, suggesting that fibrin is formed at the sites of endothelial cell PSer exposure.

To address the reviewer’s point, we have included this new experiment as a new Supplemental Figure 5 as below and stated in the results section, line 217-219.

Supplementary Figure 5. Damaged endothelial cells act as a prothrombotic surface for hemostasis.

When mechanical injury was introduced, fibrin accumulation (detected via a fibrin specific antibody) was observed at the damaged edge of endothelial cells (large arrow) adjacent to the exposed area and also the detached areas of the endothelial cell membrane (smaller arrows). The image shows the time point 10 min. Scale bar = 50 μ m.

3. line 95: “eptifibatide predominantly affects hemostatic plug formation via inhibition of platelet contraction, and therefore clot density.” Although this is stated in the text, it is not apparent in the figures provided.

We again apologize to the reviewer for the lack of clarity. We have shown that the areas of saturated fluorescence as areas of high platelet density in control blood and also showed in Supplementary video 2 that these high density platelet aggregates contract, especially against direction of flow. To address this specific point further, we conducted new experiments. Specifically, in these new experiments, small populations of platelets were stained with a dye of different fluorescence wavelength and therefore color (calcein, green), allowing individual platelets to be tracked as they integrate into the nascent clot and then over the entire time course of hemostasis. The results of these experiments are represented in the new Supplementary videos 3 and 4. In Supplementary video 3, healthy control blood is perfused and a subpopulation of platelets (stained in green) integrate into the nascent clot and as the clot contracts, those platelets are visibly displaced proximally and against the direction of flow in the vascular channel during the bleeding. In Supplementary video 4, eptifibatide-treated blood is used and labeled platelets do not displace from the position where they originally integrate into the nascent clot, which demonstrates that clot contraction is significantly attenuated in this case.

In addition, to address the reviewer’s point, we have changed that text in line 96-7 to be more precise and it now specifically reads:

“eptifibatide predominantly affects hemostatic plug formation via attenuation of platelet aggregate density and clot contraction.”

4. lines 171-175: “contraction of these aggregates was more pronounced in control blood than in eptifibatide-treated blood”. This isn’t apparent in the video.

We again apologize for the lack of clarity. Supplemental video 2 shows that, in healthy control blood, the platelet aggregates quickly formed (as evidenced by the high density of platelets shown as red) and then contract. The contraction was apparent when the platelet aggregates retract against the direction of flow. In addition, the high density of the platelets were measured as the areas with saturated fluorescence intensity (an indicator of high platelet density) as shown as Figure 2c (control n=3, eptifibatide n=4). We have since conducted new experiments, in which we directly observed single platelets moving closer to each other within the clot, demonstrating that the clots are decreasing in size (i.e., contracting) and that this phenomenon is markedly attenuate in eptifibatide-treated blood.

5. lines 176-179: “Higher magnification visualization of the bleeding site revealed clear co-localization of fibrinogen/fibrin with platelets in control conditions but not in the eptifibatide-treated condition (Fig. 2e).

Overall, these results indicate that the integrin α IIb β 3-fibrin interaction “glues” platelets together at the site of vascular injury, but that this interaction alone is not necessary for hemostasis in the microvasculature.” This isn’t apparent in the figure. In addition, co-localization of the 2 fluorophores does not necessarily imply binding of platelets to fibrin(ogen), and using fluorescent fibrinogen limits the ability to conclude which is fibrin and which is fibrinogen. Finally, quantification and an analysis of multiple injuries should be included to allow readers to draw their own conclusions.

We appreciate reviewer’s insightful comment and to address the reviewer’s concerns, we have conducted a new quantitative analysis, per the reviewer’s suggestion and have added new panel figures in Figure 2, accordingly. Specifically, we analyzed the fluorescence signal, pixel-by-pixel, of the entire site of vascular injury when hemostasis was achieved. New figure 2f and g below show the analyzed images of a representative experiment with control (f) and eptifibatide-treated blood samples (g) and the corresponding co-localization plot next to them. Each co-localization plot displays the pixel intensity of the red channel (X axis), corresponding to platelets, plotted against the pixel intensity of the green channel (Y axis), corresponding to fibrin(ogen), of each pixel in the above image. The red line the Pearson correlation coefficient line in which the perfect co-localization of the platelet and fibrin(ogen) fluorescence of all pixels would yield a $r=1.0$ or perfectly diagonal line. Note that in this example, the eptifibatide-treated sample showed a much lower Pearson correlation (i.e. flatter line in which $r \ll 1.0$), indicating that platelets and fibrin(ogen) were much less co-localized than in the control conditions. We then quantitatively compared (right) the Pearson correlation coefficients of multiple experiments ($n=10$ total, $n=5$ for each condition). Expectedly, a significant difference in platelet-fibrin(ogen) co-localization was observed between control and eptifibatide treated blood samples (t-test, $p=0.008$) and is much more pronounced in the control conditions. This data is now integrated into the revised manuscript as new Figure 2f-h.

However, we also agree with the reviewer in that a limitation of our assay, despite having the capability to enable quantitative analysis of platelet-fibrin(ogen) co-localization, is that co-localization does not, in and of itself, prove platelet binding to fibrin(ogen).

Therefore, we have modified our text in the revised manuscript to now read (with new text **bolded**):

“Overall, these results **suggest** that the integrin α IIb β 3-fibrin interaction “glues” platelets together at the

site of vascular injury” (Line 185-188)

6. lines 190-194 and figure 3: I may have missed it, but the text doesn't readily indicate the difference between Figure 3a and 3b. It also isn't stated why there are so few replicates of everything except the control? Would the conclusions hold up if there were more data?

We apologize to the reviewer for the confusion. Figures 3a and 3b are images of 2 replicates to give the reader a sense of the range of data we obtained. In light of the confusion, we have removed one of those images (i.e. removed Figure 3b) in the revised manuscript. Regarding the number of replicates experiment, for all experimental conditions, we conducted 3 to 8 replicates. We did in fact conduct many more experiments with control samples (n=47) to determine the dynamic range of our system. We opted to be thorough so we included all control data in our Figure and we do not believe that additional data would affect the conclusions as there is almost no overlap between the bleeding times in the control condition compared to the high shear, AVW3 antibody condition.

7. lines 220-225: “To assess the effect of exogenous TF in our system, vascular microchannels were pre-coated with 1 nM of TF and collagen prior to the seeding endothelial cells. We observed enhanced fibrinogen/fibrin accumulation (Supplementary Fig. 5b) at early time points with only a slight reduction in the bleeding time. These preliminary results highlight the utility of our system in assessing the synergistic relationship between coagulation factors in the hemostatic process.” This experiment is especially important. The hope for this model is that it will recapitulate in vivo events. Although the role of the intrinsic It would be a good idea to quantify the data on fibrin(ogen) accumulation and the bleeding time, with sufficient replicates.

We thank the reviewer for this suggestion and encountered several logistical issues, which were not unexpected. Overall, we attempted multiple experiments (n=5) with TF/collagen coated system. However, in 4 out of 5 experiments, the amount of fibrin(ogen) formation and accumulation was dramatic and led to pressure build-up in the vascular channel to the point of device rupture. Thus, while clot formation is certainly shortened, device rupture occurred before hemostasis could be achieved at the wound site. However, we were able to quantitatively analyze fibrin(ogen) accumulation immediately after the mechanical injury and “bleeding “ was initiated (up to 300 second before the devices rupture). The device coated with both collagen/TF showed significantly more fibrin(ogen) accumulation at 5 minutes after bleeding initiated (n=5 for each condition, p=0.042). This data is shown in the graph on the next page, which has also been added in the revised manuscript as new Supplemental Figure 6c.

REVIEWERS' COMMENTS:

Reviewer #1 (Remarks to the Author):

The authors have provided reasonable justification for the questions posed. The following are minor comments based on the latest document.

Minor suggestion:

1. In Figure 3 the data points for arterial shear rate data is way less compared to venous shear rates. To infer that bleeding time is much larger for arteriolar shear rates they need to few more data points for statistical significance.

Reviewer #2 (Remarks to the Author):

The authors have responded thoroughly to all criticisms. This paper has been substantially improved by the additions and changes as a result of reviewer comments.

Comments and critique:

1. The descriptions of the videos are very helpful, but it is still a bit difficult to identify them. In the descriptions they are numbered sequentially starting with #1 to #9, but the file names end with a jumble of numbers and letters, and in the middle have 266137x, with x going from 0 to 8.
2. Some videos would not play on my computer.
3. Some of the new data added is especially impressive, such as the videos showing the single cell imaging, with individual green platelets among the red ones.

Reviewer #3 (Remarks to the Author):

The authors of this manuscript have their hearts in the right place. Their rebuttal of the previous reviews is 23 pages long and includes multiple new figures and supplemental figures. My principle reservations are as follows:

1. Although they have tweaked their wording in the revised manuscript in response to the questions that were raised, it is still not clear to me that their device faithfully models hemostasis in either the microvasculature or the microvasculature, and that, after all, was their goal. I respectfully disagree with their statement that tissue factor (TF) has no role in achieving hemostasis in the microvasculature, and I am concerned that the conditions that they have used result in a decrease in fibrin accumulation, but not platelet accumulation when they mimic hemophilia A with an antibody to factor VIII. I think that they may be trying too hard to explain away the differences between events observed in their device and events observed in humans.
2. Unfortunately, half of the videos were blank when I played them on my MacBook. The former issue is probably a codec problem and it may be that passage through the Nature server produced the problem. Might I also suggest that the authors put a title on the opening frames of each video and, at the same time, insert a clock?

REVIEWERS' COMMENTS:

Reviewer #1 (Remarks to the Author):

The authors have provided reasonable justification for the questions posed. The following are minor comments based on the latest document.

We thank the Reviewer for the thoughtful suggestions and are encouraged to know that we have provided reasonable justification to the raised concerns.

Minor suggestion:

1. In Figure 3 the data points for arterial shear rate data is way less compared to venous shear rates. To infer that bleeding time is much larger for arteriolar shear rates they need to few more data points for statistical significance.

We appreciate the Reviewer's concern and apologize for the confusion. Regarding those specific experiments, to be as thorough as possible, we included all experiments conducted as healthy controls at venous shear rates (500 s⁻¹), which did lead to an unbalanced sample number compared to those conducted at experimental conditions. However, even though the sample numbers in the three experimental conditions are fewer compared to the control, one-way ANOVA tests indicate statistically significant differences among the bleeding times of those conditions, and the Dunn's multiple comparison test further shows that specifically, the bleeding times of the control experiments conducted at venous shear rates (500 s⁻¹) are statistically different from those conducted at arterial shear rates (2500 s⁻¹) with the presence of AVW3. To that end and to improve clarification of our data, we have modified the figure to denote that only these conditions are statistically different and that there is no statistically difference in bleeding times between the control venous condition and the other two experimental conditions (venous shear with the presence of AVW3 and arterial shear without AVW3). In addition, logistically, we are not able to complete additional arterial shear rate experiments with our system within the two week time frame given to us by the journal especially given that our university will be closed during the holiday season.

Reviewer #2 (Remarks to the Author):

The authors have responded thoroughly to all criticisms. This paper has been substantially improved by the additions and changes as a result of reviewer comments.

We thank the Reviewer for noting that we have responded to all criticisms and that the paper has been substantially improved.

Comments and critique:

1. The descriptions of the videos are very helpful, but it is still a bit difficult to identify them. In the descriptions they are numbered sequentially starting with #1 to #9, but the file names end with a jumble of numbers and letters, and in the middle have 266137x, with x going from 0 to 8.

We thank the Reviewer so much for this information. From our perspective we do not experience these issues and therefore we were not privy to the issues regarding file names. This could be an issue with the hosting site or decoding during the downloading process but regardless, we will work with the journal to ensure that the all movies are playable.

2. Some videos would not play on my computer.

Again, we thank the Reviewer for this information. We will work with the journal to ensure that all movies are repayable.

3. Some of the new data added is especially impressive, such as the videos showing the single cell imaging, with individual green platelets among the red ones.

We thank the Reviewer for this kind statement and appreciate the Reviewer's suggestions in the previous review. We are also excited to see that we could track individual platelet movement relatively easily in this new experimental method. We will further develop and conduct this single cell level analysis in future studies, especially to study clot contraction during hemostasis.

Reviewer #3 (Remarks to the Author):

The authors of this manuscript have their hearts in the right place. Their rebuttal of the previous reviews is 23 pages long and includes multiple new figures and supplemental figures.

We thank the Reviewer for acknowledging our effort to address all concerns raised by the Reviewers.

My principle reservations are as follows:

1. Although they have tweaked their wording in the revised manuscript in response to the questions that were raised, it is still not clear to me that their device faithfully models hemostasis in either the microvasculature or the microvasculature, and that, after all, was their goal. I respectfully disagree with their statement that tissue factor (TF) has no role in achieving hemostasis in the microvasculature, and I am concerned that the conditions that they have used result in a decrease in fibrin accumulation, but not

platelet accumulation when they mimic hemophilia A with an antibody to factor VIII. I think that they may be trying too hard to explain away the differences between events observed in their device and events observed in humans.

We appreciate the Reviewer's concern and agree that, in our current system, the role of TF and therefore, fibrin, may be underestimated. We do believe that while a vascular microenvironment without available TF still allows for the initiation of the hemostatic process, mainly by prompt platelet adhesion and activation coupled with prothrombotic surface of injured endothelial cells, the presence of TF would indeed augment the process and increase the mechanical stability of the nascent clot. This is consistent with our data showing increased amounts of fibrin(ogen), as the Reviewer suggests, at early time points as shown in experiments with TF-coated devices (Supplementary Figure 6). However, TF on the microchannel surface may not biologically function as identically as TF from smooth muscle cells. Therefore, incorporation of cellular expression of TF is ideal for true physiologic simulation of in vivo vasculature, and is a major aspect of our ongoing studies to further develop new iterations of our system. To the Reviewer's point, we have revised text in our discussion (line: 334-336) to read "...therefore our in vitro bleeding model includes neither smooth muscle vasoconstriction nor the effects of subendothelial tissue damage, such as smooth muscle cell-derived TF, which may, in turn, underestimate the role of fibrin."

2. Unfortunately, half of the videos were blank when I played them on my MacBook. The former issue is probably a codec problem and it may be that passage through the Nature server produced the problem. Might I also suggest that the authors put a title on the opening frames of each video and, at the same time, insert a clock?

Again, we thank the Reviewer for this information. We will work with the journal to ensure that all movies are repayable on PCs and Macs. We also add the opening title frames and time clocks for all the supplementary videos and we thank the Reviewer for the suggestion.